# FreGAN: Exploiting Frequency Components for Training GANs under Limited Data

**Mengping Yang**[†‡]    **Zhe Wang**[†‡∗]    **Ziqiu Chi**[†‡]    **Yanbing Zhang**[†‡]

[†]Department of Computer Science and Engineering, ECUST, China
[‡]Key Laboratory of Smart Manufacturing in Energy Chemical Process, ECUST, China
wangzhe@ecust.edu.cn
mengpingyang@mail.ecust.edu.cn

## Abstract

Training GANs under limited data often leads to discriminator overfitting and memorization issues, causing divergent training. Existing approaches mitigate the overfitting by employing data augmentations, model regularization, or attention mechanisms. However, they ignore the frequency bias of GANs and take poor consideration towards frequency information, especially high-frequency signals that contain rich details. To fully utilize the frequency information of limited data, this paper proposes FreGAN, which raises the model's frequency awareness and draws more attention to producing high-frequency signals, facilitating high-quality generation. In addition to exploiting both real and generated images' frequency information, we also involve the frequency signals of real images as a self-supervised constraint, which alleviates the GAN disequilibrium and encourages the generator to synthesize adequate rather than arbitrary frequency signals. Extensive results demonstrate the superiority and effectiveness of our FreGAN in ameliorating generation quality in the low-data regime (especially when training data is less than 100). Besides, FreGAN can be seamlessly applied to existing regularization and attention mechanism models to further boost the performance. [2]

## 1   Introduction

Generative adversarial networks (GANs) [10] have shown impressive achievements in synthesising plausible and photorealistic visual objects, such as image [18] [17] and video [39] generation, image inpainting [25], image translation [35] and so on. However, a prerequisite of such success is sufficient training data, which impedes applications of GANs in areas where only dozens of data are available or where it is challenging to collect massive data due to geographical, spatial, temporal, or privacy reasons. Thus developing data-efficient GANs that can generate plausible images under limited data, without compromising the quality, is necessary and meaningful.

Training GANs under limited data often leads to overfitting and instability issues [16] [14]. Specifically, when the discriminator (D) overfits to the limited training data, it simply remembers the input real images and classifies others as fake images, thus providing meaningless feedback to the generator (G), leading to divergent training and poor-quality generation. Ameliorating the synthesize quality under limited data is still an unexplored problem. Recent approaches for this problem include enlarging the training set with different data augmentations [52] [36] [51] [16] [14], regularizing the output of D with an additional constraint [37], and devising new network architectures [24]. However, existing methods are mainly developed from the perspective of data scale and model capacity, and they ignore

---

[∗]Corresponding author
[2]Our codes are available at https://github.com/kobeshegu/FreGAN_NeurIPS2022.

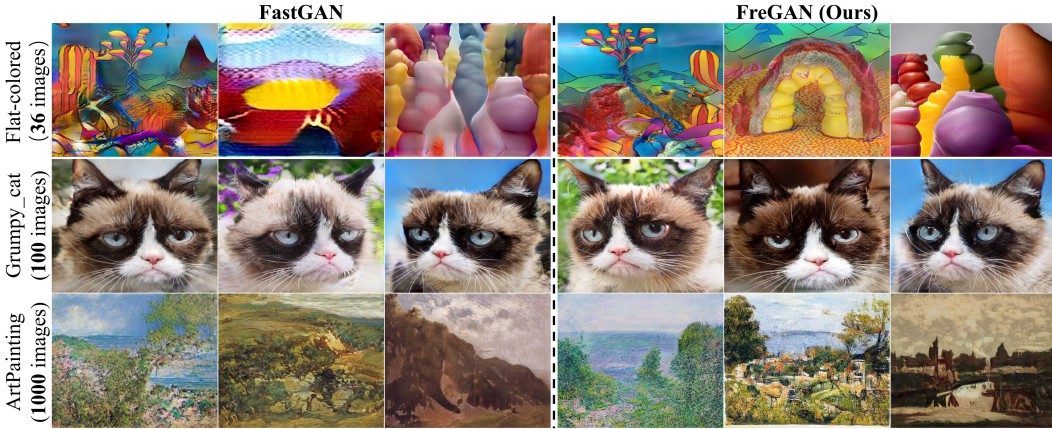

Figure 1: **Images generated by FastGAN [24] and our FreGAN given limited training data.** The details of FastGAN deteriorate while our FreGAN effectively ameliorates the synthesize quality by raising the model's frequency awareness, producing plausible images with better fine details.

a critical property of the data itself, *i.e.*, frequency signals. GANs have been demonstrated to have a spectral bias in fitting frequency signals [29] [34]. They preferentially fit low-frequency signals and tend to ignore high-frequency signals [45], which encode fine details like vertical and horizontal edges [47] [9]. Missing them may lead to unrealistic image synthesize with unsatisfactory artifacts (see Fig. 1). This paper proposes a frequency-aware model, termed as FreGAN, to raise the frequency awareness of G and D. By encouraging G to generate more reasonable and adequate high-frequency signals, our FreGAN ameliorates the synthesize quality under limited data, as shown in Fig. 1.

To fully exploit the frequency information of limited training data, we first decompose images into different frequency components via Haar wavelet transformation [6]. Unlike traditional wavelet transformation that is employed at the image level, we perform it on the intermediate features of both D and G. We then employ a high-frequency discriminator (HFD) and frequency skip connection (FSC) to raise the frequency awareness of G and D, respectively. However, G still has no explicit clue about what high-frequency signals it should synthesize, and D is overconfident in making real/fake decisions after seeing real and fake images. Such an unbalanced competition motivates us to perform high-frequency alignment (HFA) to alleviate the information asymmetry between G and D. Innovatively, we explicitly exploit the frequency signals of real images induced from D as a self-supervised constraint to guide G to leverage the frequency knowledge properly. Besides, HFD and HFA are applied on multi-scale features to thoroughly excavate the frequency signals of limited data, mitigating frequency bias and loss of high-frequency information.

The primary contributions of this paper are three-fold: 1) we propose FreGAN to raise the model's frequency awareness, which successfully mines and exploits frequency information of limited data, and as a byproduct, FreGAN alleviates the unhealthy competition between G and D; 2) we demonstrate the compatibility of our model by combining our method with other techniques like regularization [37] and attention mechanism [22]; 3) we perform extensive experiments on various datasets with limited data, and our FreGAN achieves state-of-the-art performance on these datasets, indicating the effectiveness and superiority of our method for ameliorating synthesize quality, especially when training data is extremely limited.

## 2  Related Work

**Generative Adversarial Networks.**  Generative adversarial networks (GANs) [44] [15] [11], which target at generating plausible and realistic images, have made massive progress since the pioneering work [10]. The capability of GANs enables various visual applications like image [18] [19] and video generation [39] [38], image inpainting [48] [25], image manupulation [9] [35] and super-resolution [42], etc. However, GANs are notoriously difficult to train as several issues like mode collapse and instability happen easily. Numerous techniques have been proposed to stabilize training and improve the synthesize quality by designing new optimization objectives or network architectures [13]. WGAN [2] and $f$-GAN [27] minimize the Wasserstein distance and the $f$-divergence of real and generated distribution instead of minimizing JS divergence in [10]. BigGAN [7] and

StyleGAN series [15] [18] [19] [16] [17] have made breakthrough progress in producing realistic images. However, the performance of these models deteriorates when given limited data.

**Training GANs under limited data.** Improving the synthesize quality under limited data remains an underexplored problem, which has drawn extensive attention recently. Insufficient training data leads to discriminator overfitting, thus degrading the quality of generated images. One straightforward way to address such data scarcity is to expand the training set with various augmentations. In addition to employing conventional augmentation techniques [36] [52] (e.g., flip, crop), ADA [19] and DiffAug [51] propose adaptive and differentiable augmentation to enlarge the training data, respectively. APA [14] deceives D based on the degree of overfitting with an adaptive pseudo augmentation. InsGen [46] involves instance discrimination as an auxiliary task to encourage D to distinguish every individual image, which improves the discriminative power of the discriminator. Lecam [37] regularizes the output of the discriminator throughout the training process. FastGAN [24] employs a skip-layer channel-wise excitation module and a self-supervised discriminator to stabilize and accelerate the training. The most recently MoCA [22] improves few-shot image generation quality with a prototype memory with an attention mechanism. Benefit from the significant progress of large-scale pre-trained visual recognition models, Vision-aided GAN [20] uses available off-the-shelf models to help the GAN training and ProjectedGAN [31] improve GANs by projecting generated and real images into pre-trained feature spaces. Another category of methods transfer and reuse knowledge from models that are pre-trained on large-scale data, *i.e.*, few-shot GAN adaptation [41] [23] [40] [28]. In this paper, we ameliorate the synthesize quality under limited data from the frequency domain perspective. By raising the frequency awareness of GANs and providing more fine details to G, we facilitate photorealistic image generation. Our work is complementary to previous model regularization and attention mechanism approaches, and our method promotes equilibrium between G and D.

**Wavelet Transformation in GANs.** Schwarz et al. [32] prove that GANs exhibit a frequency bias and resolving frequency artifacts is necessary for photorealistic image generation. Consequently, GANs tend to ignore high-frequency signals as they are hard to generate, compromising the generation quality. Wavelet transformation [6], which decomposes images into frequency components with different bands, has been wildly used in various applications of GANs, such as style transfer [4] [47], image inpainting [48], image editing [9], etc. HiFA [9] alleviates the generator's pressure of producing high-frequency signals by directly feeding high-frequency components to the generator. WaveFill [48] disentangles different frequency signals and explicitly fills the missing regions in each frequency band, achieving superior image inpainting. Zhang et al. [49] propose wavelet knowledge distillation towards efficient image-to-image translation without a performance drop. SWAGAN [8] incorporates wavelet with the hierarchical training of StyleGAN2 [19] and performs wavelets at the image level. Our FreGAN is more flexible by directly decomposing intermediate features of the generator and the discriminator into the wavelet domain, and no additional down/up sampling are required to convert images to higher/lower resolution as in SWAGAN, which makes our method more efficient. Unlike existing methods that are performed on ample data, this paper addresses the more challenging few-shot generation problem. In addition to raising the frequency awareness of the model, we also mitigate the unhealthy competition by lessening the frequency gap between G and D.

## 3 Methodology

The overall framework of our FreGAN is illustrated in Fig. 2. To formulate our method, we explicitly utilize wavelet transformation to decompose features into different frequency components. We then employ high-frequency discriminator (HFD) and frequency skip connection (FSC) to raise the frequency awareness of G and D, respectively. Moreover, we perform high-frequency alignment (HFA) to further guide G to synthesize adequate frequency signals.

### 3.1 Wavelet Transformation

To decompose images into different frequency components, we adopt a simple but effective wavelet transformation, *i.e.*, Haar wavelet. Haar wavelet consists of two mirror operations: wavelet pooling and wavelet unpooling. The former converts images into the hlwavelet domain, and the latter inversely reconstructs frequency components into the spatial domain. There are four kernels in wavelet pooling

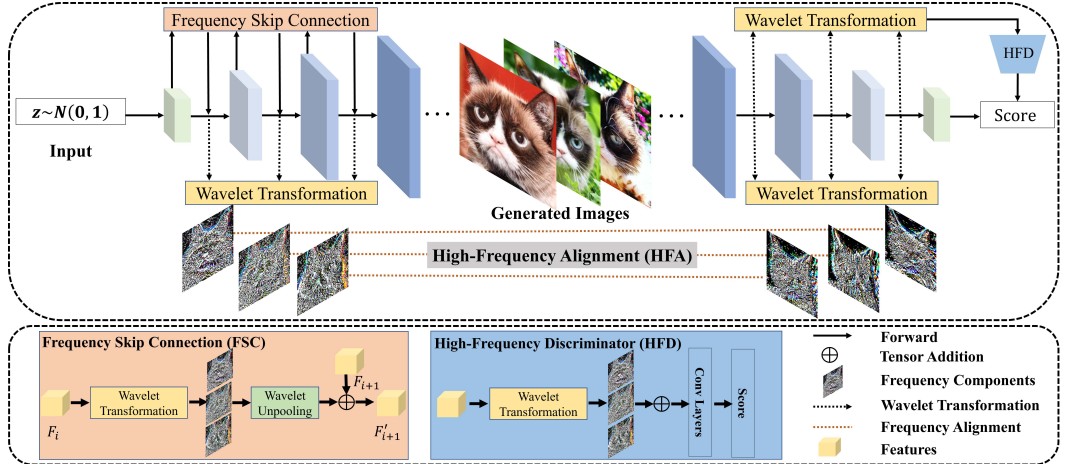

Figure 2: **The overall framework of our proposed FreGAN.** Composed of three key ingredients, *i.e.*, frequency skip connection (FSC), high-frequency discriminator (HFD), and high-frequency alignment (HFA), our FreGAN raises the model's frequency awareness, facilitating high-quality image synthesize under limited data.

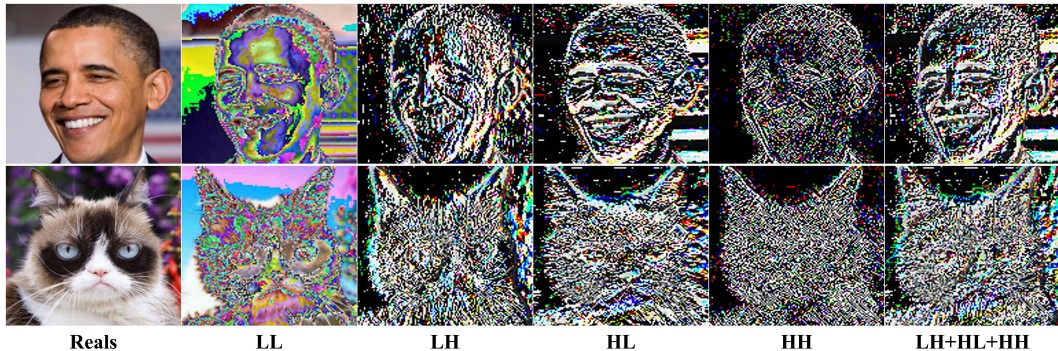

Figure 3: **Illustration of frequency components obtained from Haar wavelet transformation.** The low (L) pass filter captures images' overall textures and outlines, and the high (H) pass filter concentrates on details such as the background and edges.

operation: $LL^T, LH^T, HL^T, HH^T$, where $L^T = \frac{1}{\sqrt{2}}[1,1]$, $H^T = \frac{1}{\sqrt{2}}[-1,1]$, $L$ and $H$ denotes the low and high pass filters, respectively. The low (L) pass filter captures the outline and surface of images, while the high (H) pass filter focuses on detailed information like the edges and delicate textures. Fig. 3 illustrates the obtained frequency components of given images via Haar wavelet. We can observe that low-frequency component $LL$ contains the overall surface of images, while components that are decomposed by high pass filters, i.e., $LH, HL, HH$, contain more fine details. Further, by summing up the three high-frequency components, we approximately obtain all details information of images, *e.g.*, the eyes of the cat and the teeth of Obama.

### 3.2 High-Frequency Discriminator

To raise the frequency awareness of D, we devise high-frequency discriminator (HFD). HFD is responsible for distinguishing the real images from the generated images from the perspective of the frequency domain. Formally, for the $i$-th layer in the discriminator, we adopt wavelet pooling on the intermediate features and obtain $LL_D^i, LH_D^i, HL_D^i, HH_D^i$, then we combine the three high-frequency components by tensor addition, *i.e.*, $HF_D^i = LH_D^i + HL_D^i + HH_D^i$, which contains sufficient details of features. By applying traditional convolution and downsampling operations following the original discriminator, we define the adversarial loss of our HFD as:

$$\mathcal{L}_D^{HF} = -\mathbb{E}_{HF_{\text{real}} \sim I_{\text{real}}}[\min(0, -1 + D_H(HF_{\text{real}}))] - \mathbb{E}_{HF_{\text{fake}} \sim G(z)}[\min(0, -1 - D_H(HF_{\text{fake}})] \quad (1)$$

$$\mathcal{L}_G^{HF} = \mathbb{E}_{HF_{\text{fake}}}[D_H(HF_{\text{fake}})] \quad (2)$$

where $HF_{\text{real}}$ and $HF_{\text{fake}}$ are the high-frequency information of real and fake images, respectively. $D_H$ is the high-frequency discriminator. Since the high-frequency information may be eschewed by D as the network goes deeper, we perform multi-scale HFD on multi-layers of the discriminator. The multi-scale operation ensures fully mine and exploit the frequency information of limited data, which further improves D's frequency awareness. Notably, being guided by the HFD with Eq. 2, G is also optimized to produce rich high-frequency details.

### 3.3   Frequency Skip Connection

The generator is capable of producing plausible frequency signals after employing HFD (see Tab. 5). However, as GANs fit frequency signals from low to high and the high-frequency signals may be ignored as the network goes deeper. To prevent the loss of high-frequency information and further encourage the generator to produce rich details, we propose frequency skip connection (FSC). Concretely, we utilize wavelet unpooling operation of the frequency components $LL_G^i, LH_G^i, HL_G^i, HH_G^i$ obtained from wavelet transformation on the features of G's $i$-th layer, which reconstructs the high-frequency representation to the original features. Then we explicitly feed the reconstructed frequency representations to the next layer of G. Formally,

$$F'_{i+1} = F_{i+1} + Unpooling(LL_G^i, LH_G^i, HL_G^i, HH_G^i) \tag{3}$$

where $F_i$ denotes the features of the $i$-th layer and $Unpooling$ is the wavelet unpooling operation. $F'_{i+1}$ is the obtained features after FSC, which will be fed into the subsequent layer for further operation. Such skip connection prevents loss of high-frequency information and maintains high-frequency details.

### 3.4   High-Frequency Alignment

Adding HFD and FSC explicitly raises the frequency awareness of G, but G can only synthesize arbitrary frequency signals. How G can utilize the frequency signals is still ambiguous, and D still dominates the competition since it learns discriminative knowledge from both real and generated images. To balance the unhealthy competition between G and D, we propose high-frequency alignment (HFA), which involves high-frequency signals of real images induced from D as a regularizer to guide G, promoting G to synthesize more reasonable and realistic fine details. Specifically, we extract the frequency representations of intermediate features of G at different layers. For the $i$-th layer of G, we obtain frequency components $LL_G^i, LH_G^i, HL_G^i, HH_G^i$. We ignore $LL_G^i$ and combine the three high-frequency components, i.e., $HF_G^i = LH_G^i + HL_G^i + HH_F^i$. Then we use the high-frequency components of the discriminator $HF_D^i$ as a self-supervision constraint. In addition to fool D, G is expected to minimize the distance of high-frequency information between the generated and real images. The alignment loss is defined as:

$$\mathcal{L}_{\text{align}} = \|HF_D - HF_G\|_1 \tag{4}$$

where $\| * \|_1$ denotes the $L_1$-norm. Such alignment encourages G to synthesis frequency signals that approach real frequency signals, mitigating the unhealthy competition and facilitating generation quality. To take full advantage of frequency signals of real images from D, we perform HFA on multi-scale features like HFD as shown in Fig. 2. The ablative experiment results in Sec. 4.2 demonstrate the rationality and effectiveness of employing HFA and HFD on multi-scale features.

### 3.5   Optimization

Following [24], we adopt the hinge version of adversarial loss to train our model.

$$\mathcal{L}_D = -\mathbb{E}_{x \sim I_{\text{real}}}\left[\min(0, -1 + D(x))\right] - \mathbb{E}_{\hat{x} \sim G(z)}[\min(0, -1 - D(\hat{x}))] \tag{5}$$

$$\mathcal{L}_G = -\mathbb{E}_{z \sim \mathcal{N}}[D(G(z))] \tag{6}$$

We also use the reconstruction loss [24] to encourage the discriminator to extract more representative features.

$$\mathcal{L}_{\text{recons}} = \mathbb{E}_{\mathbf{f} \sim D_{\text{encode}}(x), x \sim I_{\text{real}}}[\|\mathcal{G}(\mathbf{f}) - \mathcal{T}(x)\|] \tag{7}$$

where $\mathbf{f}$ is the intermediate features of D, $\mathcal{G}$ and $\mathcal{T}$ denote the processing on the features $\mathbf{f}$ and the input images $x$. In sum, our discriminator is optimized by $\mathcal{L}_D, \mathcal{L}_{\text{recons}}$, and $\mathcal{L}_D^{HF}$. Our generator is optimized by $\mathcal{L}_G, \mathcal{L}_G^{HF}$ and $\mathcal{L}_{\text{align}}$, the coefficient of each loss is set to 1.

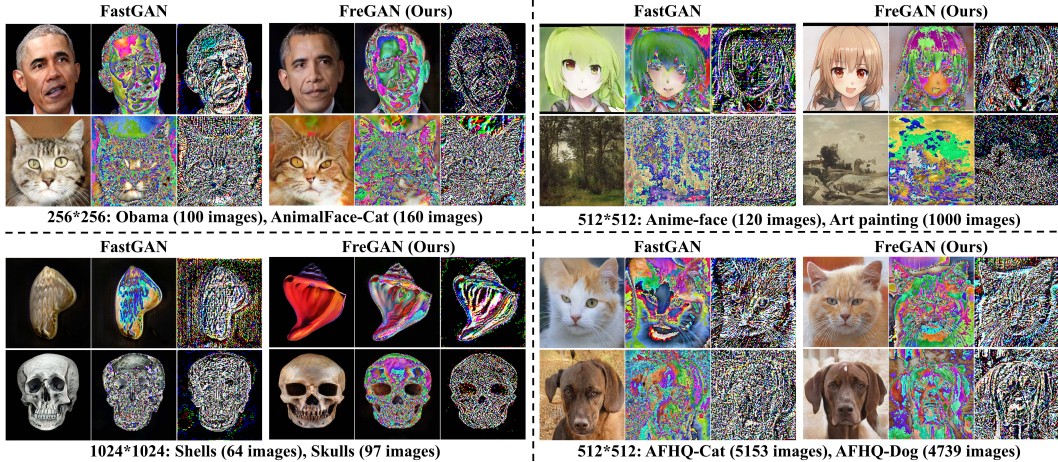

FastGAN    FreGAN (Ours)      FastGAN    FreGAN (Ours)

256*256: Obama (100 images), AnimalFace-Cat (160 images)    512*512: Anime-face (120 images), Art painting (1000 images)

FastGAN    FreGAN (Ours)      FastGAN    FreGAN (Ours)

1024*1024: Shells (64 images), Skulls (97 images)    512*512: AFHQ-Cat (5153 images), AFHQ-Dog (4739 images)

Figure 4: **Qualitative comparison results of our FreGAN and baseline FastGAN.** The images from left to right are generated images, low-frequency, and high-frequency components, respectively. Our FreGAN improves the overall quality of generated images and raises the model's frequency awareness, encouraging the generator to produce precise high-frequency signals with fine details.

## 4 Experiments

**Datasets.** We test the effectiveness of our method on low-shot datasets from various domains with different resolutions. On $256 \times 256$ resolution, we use Animal Face Dogs and Cat [33], as well as 100-shot-Panda, Obama, and Grumpy_cat [51]. On $512 \times 512$ resolution, we use Anime-Face, Art Paintings, Moongate, Flat-colored, and Fauvism-still-life [24]. On $1024 \times 1024$ resolution, we use Pokemon, Skulls, Shells, MetFace [16] and BrecaHAD [1]. These datasets contain a limited number of samples (mostly less than 1,000) and cover art paintings, realistic photos, human faces, etc. For datasets that are not strictly equal to the corresponding resolution, we resize them to the closest resolution in implementation. Besides, we use AnimalFace HQ (AFHQ) datasets [5] to evaluate the performance of our model when training with more data ($\sim$5k).

**Evaluation metrics and baseline.** We adopt two common metrics to evaluate the synthesize quality: Fréchet Inception Distance (FID) [12] and Kernel Inception Distance (KID) [3]. The lower FID and KID is, the better the generation quality is. FID quantifies the distance between the distribution of the generated and the real images. KID, which is designed unbiased, has been proven more descriptive for small datasets [16], note that all KID scores reported in our paper need to $\times 10^{-3}$ following [16]. Following [24], we calculate FID and KID by measuring the distance between all available training images and 5k generated images. We also provide the LPIPS [50], IS [30], Precision, Recall [21], Density, Coverage [26] results in the appendix.

We compare our model with: 1) the state-of-the-art generative model StyleGAN2 [19], and SWA-GAN [8], which incorporates wavelet into StyleGAN2; 2) data augmentation-based approaches that is designed for training GANs with limited data, *i.e.*, ADA [16], DiffAug [51], APA [14]; 3) the state-of-the-art few-shot generative model FastGAN [24]. We reimplement all baselines with their released official code under consistent settings for a fair comparison. Implementation details of baseline models are given in the appendix.

**Implementation details.** We choose the current state-of-the-art few-shot generative model Fast-GAN [24] as the backbone and implement our proposed techniques upon it. all other settings remain the same as [24]. We decompose the intermediate $8 \times 8$, $16 \times 16$, $32 \times 32$ features of G and D into frequency components for our frequency skip connection, high-frequency discriminator, and high-frequency alignment. More implement details are given in the appendix.

### 4.1 Main Results

**Quantitative comparison on datasets with limited data amounts.** The quantitative comparison results of our FreGAN and baseline methods on different resolutions are given in Tab. 1, Tab. 2 and Tab. 3. We save the best training snapshots of each method and generate 5k images to compute FID and KID. The whole training set is adopted as the referenced distribution. We can observe from the

Table 1: The FID (lower is better) and KID (lower is better) scores of our method compared to state-of-the-art methods on **256 × 256 datasets with limited data amounts**.

| | Animal Face | | | | 100-shot | | | | | |
| | Dog (389 imgs) | | Cat (160 imgs) | | Panda | | Obama | | Grumpy_cat | |
| Method | FID | KID | FID | KID | FID | KID | FID | KID | FID | KID |
|---|---|---|---|---|---|---|---|---|---|---|
| StyleGAN2 [19] | 113.86 | 91.31 | 79.04 | 34.43 | 18.05 | 7.40 | 69.01 | 52.63 | 35.00 | 11.01 |
| SWAGAN [8] | 82.47 | 80.46 | 59.71 | 19.35 | 27.55 | 11.92 | 71.05 | 55.82 | 38.44 | 17.03 |
| ADA [16] | 55.48 | 18.42 | 37.95 | 6.43 | 14.17 | 6.53 | 43.17 | 13.23 | 43.80 | 45.01 |
| APA [14] | 81.16 | 26.42 | 42.60 | 7.97 | 19.21 | 10.80 | 42.97 | 15.71 | 28.10 | 5.53 |
| DiffAug [51] | 61.34 | 24.51 | 41.84 | 12.27 | 11.52 | 3.57 | 48.85 | 23.31 | 26.89 | 9.42 |
| FastGAN [24] | 52.46 | 18.22 | 33.85 | 4.99 | 9.70 | 1.60 | 35.80 | 5.50 | 25.75 | **3.41** |
| FreGAN (Ours) | **47.85** | **13.49** | **31.05** | **2.44** | **8.97** | **0.91** | **33.39** | **3.76** | **24.93** | 3.89 |

Table 2: The FID (lower is better) and KID (lower is better) scores of our method compared to state-of-the-art methods on **512 × 512 datasets with limited data amounts**.

| | AnimeFace | | ArtPainting | | Moongate | | Flat | | Fauvism | |
| | 120 imgs | | 1000 imgs | | 136 imgs | | 36 imgs | | 124 imgs | |
| Method | FID | KID | FID | KID | FID | KID | FID | KID | FID | KID |
|---|---|---|---|---|---|---|---|---|---|---|
| StyleGAN2 [19] | 183.44 | 242.83 | 100.35 | 113.75 | 288.25 | 93.14 | 285.61 | 214.47 | 299.15 | 220.14 |
| SWAGAN [8] | 189.71 | 216.39 | 56.95 | 22.50 | 302.72 | 99.47 | 293.94 | 232.53 | 291.66 | 226.21 |
| ADA [16] | 59.67 | 16.02 | 46.38 | 12.26 | 149.06 | 43.21 | 248.46 | 62.89 | 201.99 | 86.64 |
| APA [14] | 58.38 | 15.73 | 47.23 | 10.60 | 193.67 | 50.52 | 233.52 | 166.53 | 197.47 | 66.13 |
| DiffAug [51] | 135.85 | 148.51 | 49.25 | 18.42 | 136.12 | 48.04 | 340.14 | 247.41 | 223.58 | 117.10 |
| FastGAN [24] | 55.87 | 11.17 | 45.06 | 10.26 | 114.79 | 23.57 | 216.27 | 36.88 | 178.42 | 58.01 |
| FreGAN (Ours) | **50.19** | **4.58** | **43.13** | **9.71** | **107.13** | **15.58** | **178.10** | **18.35** | **171.95** | **49.81** |

Table 3: The FID (lower is better) and KID (lower is better) scores of our method compared to state-of-the-art methods on **1024 × 1024 datasets with limited data amounts**.

| | Shells | | Skulls | | Pokemon | | BrecaHAD | | MetFace | |
| | 64 imgs | | 97 imgs | | 833 imgs | | 162 imgs | | 1336 imgs | |
| Method | FID | KID | FID | KID | FID | KID | FID | KID | FID | KID |
|---|---|---|---|---|---|---|---|---|---|---|
| StyleGAN2 [19] | 133.31 | 33.36 | 234.54 | 209.22 | 161.28 | 161.98 | 174.07 | 176.32 | 66.97 | 55.53 |
| SWAGAN [8] | 185.96 | 85.25 | 203.49 | 178.96 | 80.94 | 68.02 | 162.53 | 119.64 | 31.56 | 13.96 |
| ADA [16] | 133.22 | 29.12 | 97.05 | 12.33 | 66.41 | - | 76.67 | 21.38 | 24.74 | 10.23 |
| APA [14] | 136.52 | 58.77 | 99.46 | 12.74 | 51.05 | 59.29 | 75.89 | 25.08 | 26.03 | **5.58** |
| DiffAug [51] | 151.94 | 54.73 | 124.23 | 38.12 | 62.73 | 50.68 | 93.71 | 31.62 | 27.45 | 11.55 |
| FastGAN [24] | 141.71 | 37.00 | 101.94 | 12.10 | 44.96 | 17.31 | 59.80 | 7.24 | 26.80 | 7.08 |
| FreGAN (Ours) | **125.77** | **20.58** | **86.12** | **5.47** | **38.88** | **10.42** | **54.88** | **3.41** | **25.42** | 5.93 |

results that, although evaluated on various datasets that have different resolutions and data amounts, our proposed FreGAN achieves superior performance on all these datasets. Our FreGAN consistently improves both FID and KID metrics on 14 of the 15 datasets, demonstrating the effectiveness and generalizability of our proposed techniques. Notably, for those datasets with extremely limited data (less than 100), *i.e.*, Flat (Tab. 2), Shells and Skulls (Tab. 3), our method improves the performance more significantly, *e.g.*, the FID from 216.27 to **178.10** on Flat and from 101.94 to **86.12** on Skulls, and the corresponding KID is improved doubled, further reflecting our model's potential for training GANs with extremely limited data. More quantitative results are presented in the appendix.

**Qualitative Comparison.** The qualitative results of FastGAN and our FreGAN on various datasets are illustrated in Fig. 4. For each dataset in Fig. 4, from left to right are generated images, the visualization of the low and high-frequency components of the generated images. The images generated by FastGAN contain unsatisfactory artifacts and some of them are incongruous, *e.g.*, the generated images of cat and dog in the bottom right of Fig.4, the cat has artifacts around the head, and the dog's ears are distorted. Our FreGAN significantly facilitates image quality in coordination, rationality, and fine details. As can been seen from Fig. 4, the human face Obama generated by our FreGAN is more photorealistic, the details of the anime face, such as eye color and hair texture, are more realistic, and the synthesized animal faces of cats and dogs are also more plausible. Besides, the frequency components of the images generated by our FreGAN contain wealthier details. For example, the generated image of AnimalFace-Cat has a richer background, and the generated image of Skulls has more clear contours of the eye and nose. Such observation reflects that the proposed FreGAN: 1) ameliorates the quality of generated images under limited data; 2) raises frequency awareness of synthesizing high-frequency signals with richer fine details of images; and 3) takes full advantage of limited data's frequency information. More qualitative results are given in the appendix.

Table 4: The FID (lower is better) and KID (lower is better) scores of our method compared to the state-of-the-art FastGAN on **AFHQ [5] datasets with more training data ($\sim$5k)**.

| Method | AFHQ-Cat (5153 imgs) | | AFHQ-Dog (4739 imgs) | | AFHQ-Wild (4738 imgs) | |
| --- | --- | --- | --- | --- | --- | --- |
| | FID | KID | FID | KID | FID | KID |
| FastGAN [24] | 10.17 | 4.91 | 25.36 | 14.29 | 7.30 | 1.93 |
| +Ours | **6.62** | **1.95** | **20.75** | **11.45** | **6.37** | **1.31** |

Table 5: **Ablation studies on different components of our FreGAN.** We remove each component to evaluate the efficacy of the three ingredients of our method, *i.e.*, HFD, HFA, and FSC. The "Full" represents the the full version combining all three techniques used in the main experiments.

| Module | 100-shot-Obama ($256 \times 256$) | | Anime Face ($512 \times 512$) | | Pokemon ($1024 \times 1024$) | |
| --- | --- | --- | --- | --- | --- | --- |
| | FID | KID | FID | KID | FID | KID |
| Baseline | 35.80 | 5.50 | 55.87 | 11.17 | 44.96 | 17.31 |
| w/o HFD | 35.67 | 7.78 | 55.17 | 8.16 | 41.75 | 13.69 |
| w/o HFA | 34.28 | 4.60 | 54.40 | 10.70 | 40.27 | 12.53 |
| w/o FSC | 33.52 | 4.18 | 51.15 | 4.83 | 39.41 | 11.13 |
| Full | **33.39** | **3.76** | **50.19** | **4.58** | **38.88** | **10.42** |

**Effectiveness under datasets with more data.** To investigate the effectiveness of our FreGAN more comprehensively, we evaluate the performance on datasets with more training data, *i.e.*, AnimalFace-HQ (AFHQ) [5], which includes 3 sub-datasets with close to 5k images, the results are shown in Tab. 4. Similarly, our method yields compelling improvements on both FID and KID metrics when training with more data. Combined with the generated images in Fig. 4, the results further validate our FreGAN's contribution to the synthesize quality. Our method boosts the performance under different amounts of data, suggesting the generalization of our model.

## 4.2 Ablation Studies

**Ablation studies on variants of FreGAN.** There are three ingredients of our FreGAN, *i.e.*, the high-frequency discriminator (HFD), high-frequency alignments (HFA), and frequency skip connection (FSC). We evaluate the efficacy of each component by removing each of them from the full version of our FreGAN. We choose one from each of the different resolution datasets, *i.e.*, 100-shot-Obama, Anime face and pokemon for 256, 512 and 1024 resolution, respectively. As shown in Tab. 5, removing any of the three techniques leads to a performance drop, reflecting the contribution of each component. Still, all these variants outperform baseline FastGAN on both FID and KID, which implies that the combination of different components of our method consistently boosts model performance. Moreover, the performance drops the most when removing the HFD module, which is reasonable because the HFD raises the frequency awareness of G and D, and the frequency awareness of D serves as a self-supervision to guide G to synthesize adequate and reasonable frequency signals. Qualitative comparison results of ablation studies are given in the appendix.

**Ablation studies on different scale of features.** We employ our proposed HFD and HFA on multi-scale features of G and D, namely, 8, 16, and 32 scales of features. Here we provide the ablation studies on different scales in Tab. 6. It can be seen that performing HFD and HFA on multi-scale features boosts the model performance. Besides, when only performing HFD and HFA on single-scale features, the obtained results still outperform the FastGAN baseline, suggesting the effectiveness of HFA and HFD. Notably, despite adding more scales of features may bring further performance advancement, the required additional convolutional and downsampling layers increases for higher scales features(*e.g.*, 128, 256), bringing non-negligible computational costs.

**Ablation studies on different frequency components.** Three high-frequency components are obtained from wavelet transformation on the features, *i.e.*, $LH$, $HL$, and $HH$. Each of them encodes different details of features as shown in Fig. 3, we sum them to fuse all the detail information for further operation in our main experiments. Here we conduct experiments on the three components respectively to verify their contribution and the necessity of fusing them. As shown in Tab. 6, each high-frequency component contributes to the model performance compared with the baseline, and fusing them can better promote the generation quality.

Table 6: **Ablation studies on the different scales of HFD and HFA on AnimalFace-Dog dataset.** The numbers after "Feat" indicate the scale of features. "LH" and "HL" denote the high-frequency components. The "Full" represents the full version of our FreGAN used in the main experiments.

| Module | Metric | Baseline | Feat8 | Feat8 + 16 | LH | LH + HL | Full |
|--------|--------|----------|-------|------------|-----|---------|------|
| HFD | FID | 52.46 | 51.86 | 50.63 | 52.15 | 51.79 | **47.85** |
| HFD | KID | 18.22 | 17.13 | 16.28 | 17.94 | 16.57 | **13.49** |
| HFA | FID | 52.46 | 52.23 | 51.93 | 51.60 | 49.71 | **47.85** |
| HFA | KID | 18.22 | 17.15 | 16.87 | 16.92 | 15.87 | **13.49** |

Table 7: The FID (lower is better) and KID (lower is better) scores of our method **combined with model regularization and attention mechanism techniques on $256 \times 256$ datasets.**

| | Animal Face | | | | 100-shot | | | | | |
|--------|-----|-----|-----|-----|-----|-----|-----|-----|-----|-----|
| | Dog (389 imgs) | | Cat (160 imgs) | | Panda | | Obama | | Grumpy_cat | |
| Method | FID | KID | FID | KID | FID | KID | FID | KID | FID | KID |
| Lecam [37] | 54.88 | - | 34.18 | - | 10.16 | - | 33.16 | - | 24.93 | - |
| + Ours | **48.29** | **14.16** | **31.77** | **2.22** | **8.87** | **1.06** | **32.69** | **4.99** | **24.39** | **2.36** |
| MoCA [22] | 54.04 | 19.25 | 38.04 | 8.40 | 11.24 | 4.00 | 42.26 | 17.03 | 25.59 | 4.20 |
| + Ours | **50.96** | **16.06** | **35.47** | **4.92** | **9.05** | **1.13** | **34.13** | **5.53** | **24.78** | **3.11** |

## 4.3 Analysis on Compatibility and GAN Equilibrium

**Compatibility of Our Model.** Lecam [37] and MoCA [22] exploit regularization and attention mechanism for training GANs under limited data, respectively. We implement our proposed techniques on them to test the compatibility of our method. We keep the original setting unchanged and the set the regularization weight to 0.1. The FID results are given in Tab. 7, from which we can see that FreGAN can further boost the performance of MoCA and Lecam, demonstrating that our method is complementary to the model regularization and attention mechanism methods.

**GAN Equilibrium is improved.** Our HFA module aligns the frequency components of real and generated images, guiding G to synthesize precise instead of arbitrary high-frequency signals. Meanwhile, as a byproduct, the HFA mitigates the domain gap between G and D, alleviating the unhealthy competition. As shown in Fig. 5 (a), our discriminator converges to a better point, and our generator can better fool the discriminator, while the discriminator of FastGAN surpass the generator, thus providing less informative guidelines and degrading the synthesize quality. Besides, we plot the FID and KID curves throughout the training process in Fig. 5 (b), from which we can observe that our FreGAN are consistently better. Moreover, we plot the multi-scale HFA loss curves in Fig. 5 (c), where each line denotes the loss of each scale. These curves indicate that the frequency signals are well aligned, lessening the domain gaps and promoting the GAN equilibrium.

## 5 Discussion

**Conclusions.** In this paper, we propose a frequency-aware method for training GANs under limited data, *i.e.*, FreGAN. The proposed FreGAN ameliorates the synthesize quality by raising the model's frequency awareness, encouraging the model to pay more attention to frequency signals, especially high-frequency signals, which encode fine details of images. We conduct extensive experiments on various datasets with different amounts of data and different resolutions to demonstrate the efficacy of our proposed method. Qualitative results suggest that our model successfully makes the generator to generate precise high-frequency signals, facilitating high-quality image generation. Quantitative results indicate that our method 1) substantially boosts the performance, especially when data is

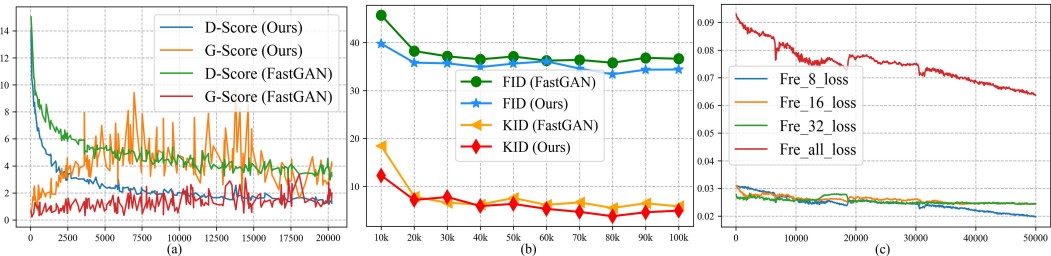

Figure 5: **Equilibrium is improved with our proposed techniques.** (a) Our generator can better deceive the discriminator. (b) Our FreGAN yields better performance throughout the training process. (c) The aligned HFA loss mitigates the asymmetrical information and facilitates the GAN equilibrium.

extremely limited (less than 100), and 2) is complementary to existing regularization and attention models. Moreover, the proposed model alleviates the disequilibrium of GANs by lessening the frequency information gap. In the future, we plan to implement our techniques on more backbones, *e.g.*, StyleGAN2 [19] and apply our method to more applications.

**Limitations.** Despite achieving significant improvements on various low-data datasets, our FreGAN still struggles in generating photorealistic images when given datasets with limited data but various contents, *e.g.*, only dozens of images, and their contents vary widely. When the low-data datasets are imbalanced [43] or even long-tailed, the proposed method may fail to generalize, which is limited by the intrinsic reasons of the data distribution. Developing more effective ways to train generative models with insufficient training data still requires more efforts.

## Acknowledgment

This work is supported by Shanghai Science and Technology Program "Distributed and generative few-shot algorithm and theory research" under Grant No. 20511100600 and "Federated based cross-domain and cross-task incremental learning" under Grant No. 21511100800, Natural Science Foundation of China under Grant No. 62076094, Chinese Defense Program of Science and Technology under Grant No.2021-JCJQ-JJ-0041, China Aerospace Science and Technology Corporation Industry-University-Research Cooperation Foundation of the Eighth Research Institute under Grant No.SAST2021-007.

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
