# Supplementary Material for "FreGAN: Exploiting Frequency Components for Training GANs under Limited Data"

**Mengping Yang**[†‡]    **Zhe Wang**[†‡*]  **Ziqiu Chi**[†‡]    **Yanbing Zhang**[†‡]
[†]Department of Computer Science and Engineering, ECUST, China
[‡]Key Laboratory of Smart Manufacturing in Energy Chemical Process, ECUST, China
`wangzhe@ecust.edu.cn`
`mengpingyang@mail.ecust.edu.cn`

## A    Appendix

In this appendix, we first provide the broader impact of our method. Then we provide more implement details (Sec. A.1), quantitative results of other metrics (Sec. A.2 and Sec. A.3), and qualitative results of generated images and ablation studies (Sec. B.1) that is not elaborated in the main paper. We also provide the quantitative comparison results of our FreGAN and FastGAN on large-scale datasets in Sec.refsec:largedatasets Besides, we provide the latent space interpolation and nearest neighbors found from training images in Sec. C for a more comprehensive analysis on the diversity of our generated images. Lastly, more quantitative and qualitative comparison results of spectral properties are given in Sec. D, demonstrating that our FreGAN is frequency-aware and can indeed produce realistic frequency signals.

**Broader impact.** The proposed method enables data-efficient GANs training for high-quality image synthesize with limited data. It benefits the practical implementation if GANs in various applications that without sufficient training samples, *e.g.*, medical images and art paintings. And our analysis of the effectiveness of frequency components in image synthesis may also extend the breadth and potential of approaches for training effective GANs from the frequency domain perspective. Besides, being capable of generating plausible and photorealistic images, our method bring potential issue of image abuse and fraud with the generated fake images. However, we believe that the rational use of such advanced technology can bring benefits to more fields like films and art production.

### A.1   More implement details.

**Implement details of our FreGAN.**   We perform Haar wavelet transformation on the intermediate $8 \times 8$, $16 \times 16$, $32 \times 32$ features of G and D. The PyTorch-like pseudocode of Haar wavelet transformation is given in Algorithm. 1. We perform FSC by wavelet unpooling the decomposed high-frequency components and feeding the reconstructed features to the subsequent layers. For HFD, we aggregate the high-frequency components by adding $LH, HL, HH$ and then employ additional downsampling and convolutional layers to compute the output scores. Specifically, the architecture resembles the original discriminator. The added layers on the high-frequency components include 2d convolutional, 2d Batch Normalization, and LeakyReLU layers. The difference is that the HFD discriminates the input images' frequency information, raising the discriminator's frequency awareness. For HFA, we align the summed frequency signals of real and generated intermediate features by minimizing Eq.4 in the main paper. Notably, only 1-2 additional layers are added for each high-frequency component without requiring much computational cost. We train our model for 100k iterations and save the checkpoints every 10k iterations. The saved checkpoints are used to generate

---

[*]Corresponding author

36th Conference on Neural Information Processing Systems (NeurIPS 2022).

**Algorithm 1:** Haar wavelet transformation pseudocode, PyTorch-like

```python
def get_wav(in_channels, pool=True):
    # wavelet decomposition using conv2d
    harr_wav_L = 1 / np.sqrt(2) * np.ones((1, 2))
    harr_wav_H = 1 / np.sqrt(2) * np.ones((1, 2))
    harr_wav_H[0, 0] = -1 * harr_wav_H[0, 0]

    harr_wav_LL = np.transpose(harr_wav_L) * harr_wav_L
    harr_wav_LH = np.transpose(harr_wav_L) * harr_wav_H
    harr_wav_HL = np.transpose(harr_wav_H) * harr_wav_L
    harr_wav_HH = np.transpose(harr_wav_H) * harr_wav_H

    filter_LL = torch.from_numpy(harr_wav_LL).unsqueeze(0)
    filter_LH = torch.from_numpy(harr_wav_LH).unsqueeze(0)
    filter_HL = torch.from_numpy(harr_wav_HL).unsqueeze(0)
    filter_HH = torch.from_numpy(harr_wav_HH).unsqueeze(0)

    if pool:
        net = nn.Conv2d
    else:
        net = nn.ConvTranspose2d
    LL = net(in_channels, in_channels*2,
            kernel_size=2, stride=2, padding=0, bias=False,
            groups=in_channels)
    LH = net(in_channels, in_channels*2,
            kernel_size=2, stride=2, padding=0, bias=False,
            groups=in_channels)
    HL = net(in_channels, in_channels*2,
            kernel_size=2, stride=2, padding=0, bias=False,
            groups=in_channels)
    HH = net(in_channels, in_channels*2,
            kernel_size=2, stride=2, padding=0, bias=False,
            groups=in_channels)

    LL.weight.requires_grad = False
    LH.weight.requires_grad = False
    HL.weight.requires_grad = False
    HH.weight.requires_grad = False

    LL.weight.data = filter_LL.float().unsqueeze(0).expand(in_channels*2, -1, -1, -1)
    LH.weight.data = filter_LH.float().unsqueeze(0).expand(in_channels*2, -1, -1, -1)
    HL.weight.data = filter_HL.float().unsqueeze(0).expand(in_channels*2, -1, -1, -1)
    HH.weight.data = filter_HH.float().unsqueeze(0).expand(in_channels*2, -1, -1, -1)

    return LL, LH, HL, HH

class WavePool(nn.Module):
    def __init__(self, in_channels):
        super(WavePool, self).__init__()
        self.LL, self.LH, self.HL, self.HH = get_wav(in_channels)

    def forward(self, x):
        return self.LL(x), self.LH(x), self.HL(x), self.HH(x)

class WaveUnpool(nn.Module):
    def __init__(self, in_channels, option_unpool='cat5'):
        super(WaveUnpool, self).__init__()
        self.in_channels = in_channels
        self.option_unpool = option_unpool
        self.LL, self.LH, self.HL, self.HH = get_wav(self.in_channels, pool=False)

    def forward(self, LL, LH, HL, HH, original=None):
        if self.option_unpool == 'sum':
            return self.LL(LL) + self.LH(LH) + self.HL(HL) + self.HH(HH)
        elif self.option_unpool == 'cat5' and original is not None:
            return torch.cat([self.LL(LL), self.LH(LH), self.HL(HL), self.HH(HH), original], dim=1)
        else:
            raise NotImplementedError
```

images for evaluation. All experiments are run on 2 Tesla V100 GPUs with PyTorch framework, and our code will be made available online.

**Implement details of baseline methods.** We reimplement all the baseline methods with their official code for fair comparisons. For StyleGAN2 [9][2], ADA [6][3], DiffAug [19][4], we keep most of the details unchanged, including style mixing regularization, path length regularization, exponential moving average of weights, non-saturating logistic loss with $R_1$ regularization. We show the discriminator for 2,000 kimg and use the best training snapshots of each model to generate 5k images for evaluation. For FastGAN [12][5], similarly, we keep all the details unchanged and implement our proposed techniques upon it, we train both our FreGAN and FastGAN for 100k iterations, and we use the saved checkpoints to generate 5k images for evaluation. All of our experiments are run on 2 Tesla V100 GPUs, using PyTorch 1.8.0, and CUDA 11.1. When combining our proposed techniques upon Lecam [17] and MoCA [11], we consistently keep all the details unchanged and add our proposed techniques to them. The coefficient of the regularization term is set as 0.1 following the original paper. Notably, we use the official code and recommended parameter of MoCA. However, we achieve 42.26 on the 100-shot-Obama dataset instead of 37.19 reported in the original paper. We infer this is caused by randomness and hardware differences. Nonetheless, our proposed method contributes to the performance and complements the attention mechanism-based method.

**Datasets description.**

- AFHQ dataset[6] dataset contain ∼5k training images of animal faces with $512 \times$ resolution. The dataset is made available under the Creative Commons BY-NC 4.0 license.

- 100-shot datasets[7] contain various contents of images, and all the datasets contain 100 training images. They are ideal for verifying the quality of the generation in low-shot scenarios.

- MetFace[8] dataset contains 1336 high-quality PNG images at $1024 \times 1024$ resolution. The dataset is made available under the Creative Commons BY-NC 2.0 license.

- BrecaHAD[9] dataset contains 162 images for breast cancer histopathological annotation and diagnosis. Its texture and content are complex, thus is suited for evaluating GANs' performance under limited data, facilitating the exploration of data-efficient GANs for downstream tasks of the medical field.

- anime-face, art-paintings, moongate, flat, fauvism, shells, skulls[10]. These datasets include 60 1000 images with different resolutions. Thus we adopt them for evaluating our model under limited data. We resize them to the closest resolution in implementation.

All of the datasets we used in this paper are open-sourced, and we use them only for academic research without any commercial purposes.

## A.2 More Quantitative comparison on datasets with limited data amounts.

We evaluate the performance of our FreGAN and baseline models on more datasets with limited data amounts in Tab. 1, namely, Medici, Temple, Bridge, and Wuzhen, all of which contain only 100 training images. The resolution of these datasets is $256 \times 256$. The FID and KID results are consistent with the results in the main paper. Our FreGAN achieves better performance compared with the baseline models, suggesting the effectiveness and generalization of our model.

## A.3 More Quantitative Comparison on Datasets with limited data amounts.

In addition to the two common-used metrics, *i.e.*, FID and KID, we also compute the Precision, Recall [10], Density, Coverage [14], Inception Score (IS) [15], and LPIPS [18] results in Tab. 2,

---

[2]`https://github.com/NVlabs/stylegan2`

[3]`https://github.com/NVlabs/stylegan2-ada`

[4]`https://github.com/mit-han-lab/data-efficient-gans`

[5]`https://github.com/odegeasslbc/FastGAN-pytorch`

[6]`https://github.com/clovaai/stargan-v2`

[7]`https://data-efficient-gans.mit.edu/datasets/`

[8]`https://github.com/NVlabs/metfaces-dataset`

[9]`https://figshare.com/articles/dataset/BreCaHAD_A_Dataset_for_Breast_Cancer_Histopathological_Annotation_and_Diagnosis/7379186`

[10]`https://drive.google.com/file/d/1aAJCZbXNHyraJ6Mi13dSbe7pTyfPXha0/view/`

**FastGAN** **FreGAN (Our)**

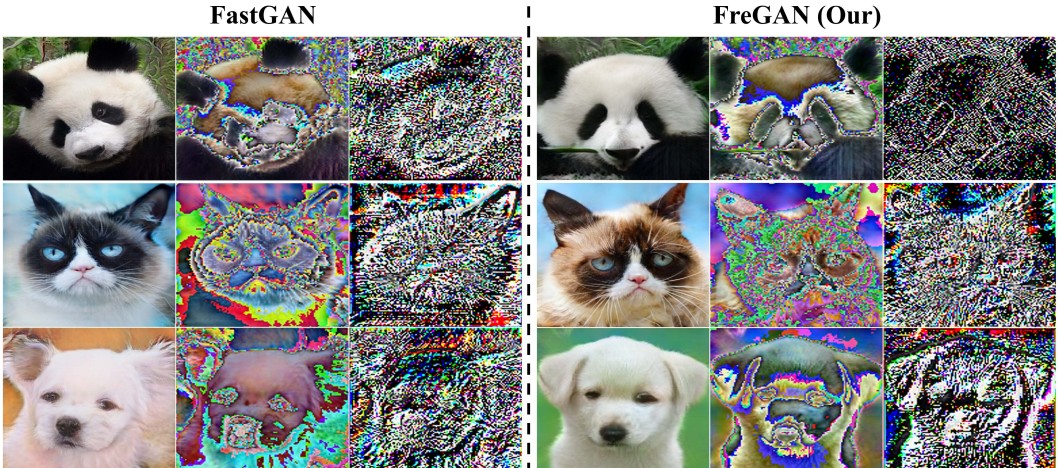

**256*256: Panda, Grumpy_cat (100 images), AnimalFace-Dog (389 images)**

Figure 1: **Qualitative comparison results of our FreGAN and baseline FastGAN.** The images from left to right are generated images, low-frequency, and high-frequency components, respectively. Our FreGAN improves the overall quality of generated images and raises the model's frequency awareness, encouraging the generator to produce precise high-frequency signals with fine details.

**FastGAN** **FreGAN (Our)**

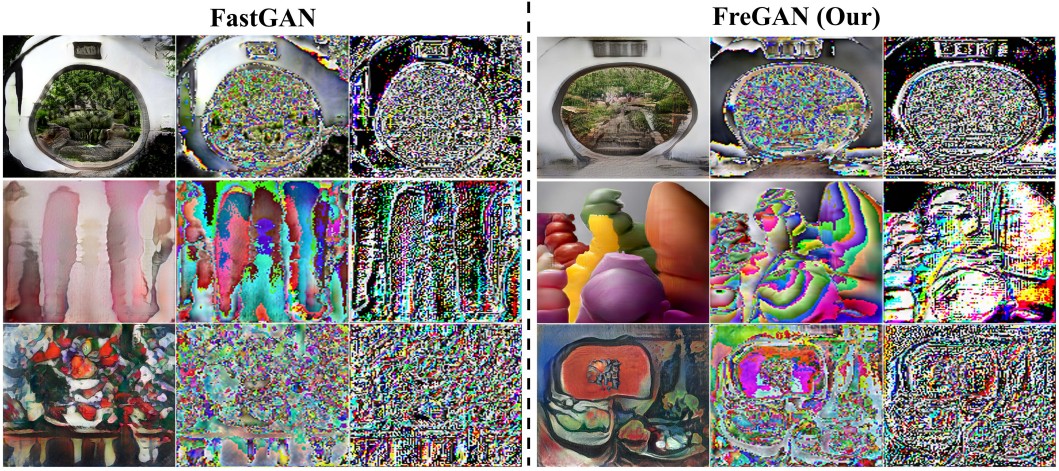

**512*512: Moongate (136 images), Flat (36 images), Fauvism (124 images)**

Figure 2: **Qualitative comparison results of our FreGAN and baseline FastGAN.** The images from left to right are generated images, low-frequency, and high-frequency components, respectively. Our FreGAN improves the overall quality of generated images and raises the model's frequency awareness, encouraging the generator to produce precise high-frequency signals with fine details.

Table 1: The FID (lower is better) and KID (lower is better) scores of our method compared to state-of-the-art methods on **256 × 256 datasets with limited data amounts**.

| Method | Medici (100 imgs) | | Temple (100 imgs) | | Bridge (100 imgs) | | Wuzhen (100 imgs) | |
|---|---|---|---|---|---|---|---|---|
| | FID | KID | FID | KID | FID | KID | FID | KID |
| StyleGAN2 [9] | 66.36 | 41.16 | 73.35 | 46.76 | 116.40 | 71.51 | 135.40 | 116.46 |
| ADA [6] | 44.21 | - | 49.72 | - | 72.07 | - | 92.81 | 47.01 |
| APA [5] | 76.11 | - | 41.38 | 10.33 | 189.74 | 98.45 | 102.10 | 45.92 |
| DiffAug [19] | 42.63 | 21.23 | 50.73 | 11.19 | 49.97 | 11.92 | 122.44 | 78.08 |
| FastGAN [12] | 38.47 | 12.63 | 36.01 | 4.95 | 46.82 | 8.25 | 67.98 | 15.13 |
| FreGAN (Ours) | **27.30** | **3.34** | **33.38** | **3.23** | **44.18** | **7.25** | **59.89** | **6.62** |

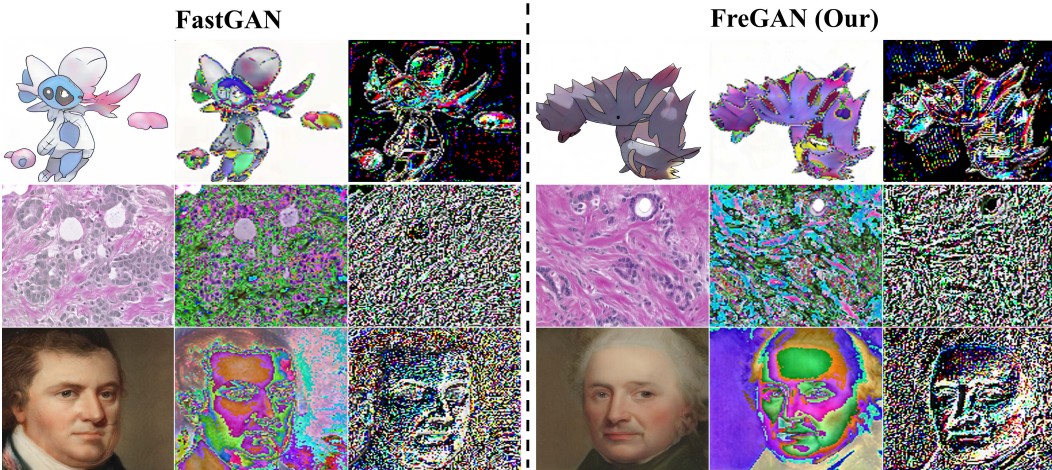

**FastGAN**      **FreGAN (Our)**

**1024\*1024: Pokemon (833 images), BrecaHAD (162 images), MetFace (1336 images)**

Figure 3: **Qualitative comparison results of our FreGAN and baseline FastGAN.** The images from left to right are generated images, low-frequency, and high-frequency components, respectively. Our FreGAN improves the overall quality of generated images and raises the model's frequency awareness, encouraging the generator to produce precise high-frequency signals with fine details.

Tab. 3, Tab. 4, Tab. 5, Tab. 6, Tab. 7, Tab. 8, Tab. 9, Tab. 10, Tab. 11, Tab. 12, Tab. 13, Tab. 14, Tab. 15, Tab. 16, Tab. 17, Tab. 18, Tab. 19, Tab. 20.

Precision is quantified by calculating if the generated images are within the estimated manifold of real images, and symmetrically, Recall is quantified by calculating if the real images are within the estimated manifold of generated images [10]. Precision evaluates the probability of generated images falling into the real distribution, and Recall is the opposite. Density and Coverage evaluate the fidelity and diversity of generative models, respectively [14], they are claimed successfully detect two identical distributions and are not robust against outliers. Specifically, we use the generated 5k images of each model corresponding to the best FID results. The whole training images are used as the referenced distribution. We set the nearest k as 5 and compute the Precision, Recall, Density, and Coverage based on the official code of [14][11]. We split the generated into 10 parts for IS and report the mean and standard deviation of the calculated IS scores. We compute LPIPS [18][12] of all the paired of 5k images generated by each methods and report the average and standard deviation of LPIPS scores [16].

We can observe from these tables that: 1) KID and FID can consistently reflect the synthesize quality under limited data; 2) The Precision and Recall serve as supplementary metrics of evaluating generative GANs. They can reflect the distance between the generated and real distributions. However, they may be biased when training data is limited since the model tends to overfit when given limited data, the model simply replicates the training images and achieves high Precision and Recall; 3) Evaluating the fidelity of generated images, the Density metric is more consistent with FID and KID, while the Coverage is not suited for evaluating the diversity as the synthesized 5k images cover the

---

[11]`https://github.com/clovaai/generative-evaluation-prdc`
[12]`https://github.com/richzhang/PerceptualSimilarity`

Table 2: The Precision (P) (higher is better) and Recall (R) (higher is better) scores of our method compared to state-of-the-art methods on **256 × 256 datasets with limited data amounts**.

| | Animal Face | | | | 100-shot | | | | | |
| | Dog (389 imgs) | | Cat (160 imgs) | | Panda | | Obama | | Grumpy_cat | |
| Method | P | R | P | R | P | R | P | R | P | R |
|---|---|---|---|---|---|---|---|---|---|---|
| StyleGAN2 [9] | 0.33 | 0.06 | 0.40 | 0.04 | 0.61 | 0.06 | 0.47 | **0.16** | 0.74 | 0.10 |
| ADA [6] | 0.78 | **0.50** | 0.78 | 0.13 | 0.61 | 0.06 | 0.90 | 0.00 | 0.72 | 0.00 |
| APA [5] | 0.75 | 0.08 | 0.78 | 0.13 | 0.48 | 0.14 | 0.89 | 0.00 | 0.88 | 0.05 |
| DiffAug [19] | 0.79 | 0.31 | 0.87 | 1.05 | 0.79 | 0.11 | 0.85 | 0.00 | **0.97** | 0.01 |
| FastGAN [12] | **0.88** | **0.50** | 0.87 | **0.25** | **0.91** | 0.10 | 0.94 | 0.10 | 0.93 | **0.13** |
| FreGAN (Ours) | 0.86 | 0.38 | **0.90** | 0.24 | **0.91** | **0.15** | **0.96** | 0.11 | 0.94 | 0.03 |

Table 3: The Precision (P) (higher is better) and Recall (R) (higher is better) scores of our method compared to state-of-the-art methods on **256 × 256 datasets with limited data amounts**.

| | Medici (100 imgs) | | Temple (100 imgs) | | Bridge (100 imgs) | | Wuzhen (100 imgs) | |
| Method | P | R | P | R | P | R | P | R |
|---|---|---|---|---|---|---|---|---|
| StyleGAN2 [9] | 0.28 | 0.02 | 0.47 | **0.06** | 0.79 | 0.00 | 0.09 | 0.04 |
| ADA [6] | - | - | - | - | - | - | 0.50 | 0.00 |
| APA [5] | - | - | 0.91 | 0.01 | 0.25 | 0.00 | 0.44 | 0.06 |
| DiffAug [19] | 0.60 | 0.00 | 0.88 | **0.06** | 0.89 | **0.14** | 0.30 | 0.01 |
| FastGAN [12] | 0.87 | 0.00 | 0.91 | 0.05 | 0.88 | 0.09 | 0.80 | **0.15** |
| FreGAN (Ours) | **0.93** | **0.03** | **0.93** | 0.02 | **0.90** | 0.04 | **0.85** | 0.09 |

Table 4: The Precision (P) (higher is better) and Recall (R) (higher is better) scores of our method compared to state-of-the-art methods on **512 × 512 datasets with limited data amounts**.

| | AnimeFace 120 imgs | | ArtPainting 1000 imgs | | Moongate 136 imgs | | Flat 36 imgs | | Fauvism 124 imgs | |
| Method | P | R | P | R | P | R | P | R | P | R |
|---|---|---|---|---|---|---|---|---|---|---|
| StyleGAN2 [9] | 0.86 | 0.00 | 0.34 | 0.01 | 0.63 | **0.16** | 0.62 | 0.00 | 0.35 | 0.00 |
| ADA [6] | 0.89 | 0.03 | 0.71 | 0.35 | 0.54 | 0.01 | 0.75 | 0.00 | 0.76 | 0.00 |
| APA [5] | 0.91 | 0.03 | 0.67 | **0.38** | 0.46 | 0.00 | 0.50 | 0.00 | 0.73 | 0.00 |
| DiffAug [19] | 0.58 | 0.00 | 0.74 | 0.20 | 0.70 | 0.00 | 0.86 | 0.00 | 0.48 | 0.00 |
| FastGAN [12] | 0.89 | 0.12 | 0.81 | 0.32 | **0.71** | 0.02 | 0.74 | **0.03** | 0.84 | **0.02** |
| FreGAN (Ours) | **0.93** | **0.13** | **0.83** | 0.33 | **0.71** | 0.06 | **0.90** | **0.03** | 0.82 | **0.02** |

Table 5: The Precision (P) (higher is better) and Recall (R) (higher is better) scores of our method compared to state-of-the-art methods on **1024 × 1024 datasets with limited data amounts**.

| | Shells 64 imgs | | Skulls 97 imgs | | Pokemon 833 imgs | | BrecaHAD 162 imgs | | MetFace 1336 imgs | |
| Method | P | R | P | R | P | R | P | R | P | R |
|---|---|---|---|---|---|---|---|---|---|---|
| StyleGAN2 [9] | **0.73** | 0.03 | 0.12 | 0.02 | 0.69 | 0.00 | 0.53 | 0.01 | 0.74 | 0.00 |
| ADA [6] | 0.52 | 0.03 | 0.72 | 0.03 | - | - | 0.82 | 0.12 | 0.78 | 0.23 |
| APA [5] | 0.50 | 0.02 | 0.76 | 0.06 | **0.87** | 0.00 | 0.83 | 0.24 | 0.80 | 0.27 |
| DiffAug [19] | 0.56 | 0.00 | 0.57 | 0.00 | 0.69 | 0.01 | 0.68 | 0.04 | 0.82 | 0.24 |
| FastGAN [12] | 0.59 | 0.06 | 0.70 | 0.03 | 0.74 | 0.25 | **0.94** | 0.42 | **0.86** | 0.27 |
| FreGAN (Ours) | 0.65 | **0.08** | **0.83** | **0.08** | 0.80 | **0.31** | 0.94 | **0.51** | 0.86 | **0.32** |

original training images, leading high value of the Coverage, *e.g.*, equals to 1 on many datasets and baseline models; 4) Despite may be biased, our proposed FreGAN achieves advanced performance on most of the used datasets and the adopted evaluation metrics, demonstrating the effectiveness and superiority of our method; 5) Notably, the recall is very low for all datasets and methods. We infer that this is possibly due to the low diversity of the training and the fact that recall is not suitable for evaluation in scenarios with limited data; 6) Devising indicative evaluation metrics for few-shot image generation tasks remains an open and tricky problem because one has to consider the degree of overfitting, the fidelity, and the diversity.

Table 6: The Precision (P) (higher is better) and Recall (R) (higher is better) scores of our method compared to the baseline FastGAN [12] on **AFHQ (∼5k) [1] datasets with more data**.

| Method | AFHQ-Cat (5153 imgs) | | AFHQ-Dog (4739 imgs) | | AFHQ-Wild (4738 imgs) | |
| --- | --- | --- | --- | --- | --- | --- |
| | P | R | P | R | P | R |
| FastGAN [12] | 0.81 | 0.31 | 0.86 | 0.56 | 0.76 | **0.22** |
| +Ours | **0.82** | **0.45** | **0.87** | **0.69** | **0.77** | 0.21 |

Table 7: The Density (D) (higher is better) and Coverage (C) (higher is better) scores of our method compared to state-of-the-art methods on **256 × 256 datasets with limited data amounts**.

| | Animal Face | | | | 100-shot | | | | | |
| --- | --- | --- | --- | --- | --- | --- | --- | --- | --- | --- |
| | Dog (389 imgs) | | Cat (160 imgs) | | Panda | | Obama | | Grumpy_cat | |
| Method | D | C | D | C | D | C | D | C | D | C |
| StyleGAN2 [9] | 0.19 | 0.51 | 0.28 | 0.89 | 0.06 | 0.99 | 0.26 | 0.98 | 0.59 | 0.99 |
| ADA [6] | 0.61 | 0.96 | 0.85 | **1.00** | 0.06 | **1.00** | 1.23 | **1.00** | 0.30 | 0.45 |
| APA [5] | 0.62 | 0.74 | 0.78 | 0.99 | 0.14 | 0.91 | 0.97 | **1.00** | 0.95 | **1.00** |
| DiffAug [19] | 0.65 | 0.91 | 1.13 | **1.00** | 0.11 | **1.00** | 0.68 | **1.00** | **1.37** | **1.00** |
| FastGAN [12] | **0.87** | 0.96 | 1.06 | **1.00** | 0.10 | **1.00** | 1.28 | **1.00** | 1.30 | **1.00** |
| FreGAN (Ours) | 0.86 | **0.98** | **1.24** | **1.00** | **0.15** | **1.00** | **1.38** | **1.00** | 1.28 | **1.00** |

Table 8: The Density (D) (higher is better) and Coverage (C) (higher is better) scores of our method compared to state-of-the-art methods on **256 × 256 datasets with limited data amounts**.

| | Medici (100 imgs) | | Temple (100 imgs) | | Bridge (100 imgs) | | Wuzhen (100 imgs) | |
| --- | --- | --- | --- | --- | --- | --- | --- | --- |
| Method | D | C | D | C | D | C | D | C |
| StyleGAN2 [9] | 0.11 | 0.79 | 0.31 | 0.81 | 0.64 | 0.55 | 0.04 | 0.67 |
| ADA [6] | - | - | - | - | - | - | 0.27 | 0.90 |
| APA [5] | - | - | **1.27** | 0.98 | 0.06 | 0.23 | 0.29 | 0.96 |
| DiffAug [19] | 0.35 | 0.86 | 0.91 | **1.00** | 1.03 | **1.00** | 0.14 | 0.83 |
| FastGAN [12] | 0.88 | 0.98 | 1.19 | **1.00** | 1.04 | **1.00** | 0.94 | **1.00** |
| FreGAN (Ours) | **1.00** | **1.00** | 1.24 | **1.00** | **1.16** | **1.00** | **1.20** | **1.00** |

Table 9: The Density (D) (higher is better) and Coverage (C) (higher is better) scores of our method compared to state-of-the-art methods on **512 × 512 datasets with limited data amounts**.

| | AnimeFace 120 imgs | | ArtPainting 1000 imgs | | Moongate 136 imgs | | Flat 36 imgs | | Fauvism 124 imgs | |
| --- | --- | --- | --- | --- | --- | --- | --- | --- | --- | --- |
| Method | D | C | D | C | D | C | D | C | D | C |
| StyleGAN2 [9] | 0.25 | 0.10 | 0.16 | 0.34 | 0.19 | 0.38 | 0.36 | 0.67 | 0.16 | 0.27 |
| ADA [6] | 1.58 | 0.99 | 0.93 | 0.91 | 0.45 | 0.99 | 0.90 | 0.83 | 0.99 | 0.90 |
| APA [5] | **2.15** | **1.00** | 0.72 | 0.90 | 0.42 | 0.96 | 0.37 | 0.67 | 1.13 | 0.93 |
| DiffAug [19] | 0.22 | 0.30 | 0.90 | 0.87 | 0.86 | 0.90 | 0.93 | 0.56 | 0.32 | 0.75 |
| FastGAN [12] | 1.27 | **1.00** | 1.17 | 0.95 | 0.93 | **1.00** | 0.74 | 0.97 | 1.53 | 0.99 |
| FreGAN (Ours) | 1.65 | **1.00** | **1.23** | **0.97** | **1.17** | **1.00** | **0.94** | **1.00** | 1.33 | **1.00** |

Table 10: The Density (D) (higher is better) and Coverage (C) (higher is better) scores of our method compared to state-of-the-art methods on **1024 × 1024 datasets with limited data amounts**.

| | Shells 64 imgs | | Skulls 97 imgs | | Pokemon 833 imgs | | BrecaHAD 162 imgs | | MetFace 1336 imgs | |
| --- | --- | --- | --- | --- | --- | --- | --- | --- | --- | --- |
| Method | D | C | D | C | D | C | D | C | D | C |
| StyleGAN2 [9] | **0.97** | **1.00** | 0.11 | 0.71 | 0.39 | 0.13 | 0.23 | 0.47 | 0.61 | 0.51 |
| ADA [6] | 0.43 | **1.00** | 0.86 | **1.00** | - | - | 0.75 | **1.00** | 0.93 | 0.96 |
| APA [5] | 0.35 | 0.89 | 0.91 | **1.00** | **1.22** | 0.71 | 0.68 | **1.00** | 1.12 | 0.96 |
| DiffAug [19] | 0.47 | 0.94 | 0.58 | 0.99 | 0.50 | 0.63 | 0.52 | 0.96 | 1.08 | 0.96 |
| FastGAN [12] | 0.66 | 0.94 | 1.34 | **1.00** | 0.92 | 0.96 | **1.20** | **1.00** | **1.37** | 0.96 |
| FreGAN (Ours) | 0.78 | **1.00** | **1.35** | **1.00** | 1.10 | **0.97** | 1.08 | **1.00** | 1.27 | **0.97** |

Table 11: The Density (D) (higher is better) and Coverage (C) (higher is better) scores of our method compared to the baseline FastGAN [12] on **AFHQ (∼5k) [1] datasets with more data**.

| Method | AFHQ-Cat (5153 imgs) | | AFHQ-Dog (4739 imgs) | | AFHQ-Wild (4738 imgs) | |
|---|---|---|---|---|---|---|
| | D | C | D | C | D | C |
| FastGAN [12] | **1.19** | 0.80 | **0.82** | 0.50 | 1.21 | 0.72 |
| +Ours | 1.18 | **0.85** | 0.72 | **0.57** | **1.24** | **0.73** |

Table 12: The IS (higher is better) scores of our method compared to state-of-the-art methods on **256 × 256 datasets with limited data amounts**.

| | Animal Face | | 100-shot | | |
|---|---|---|---|---|---|
| | Dog (389 imgs) | Cat (160 imgs) | Panda | Obama | Grumpy_cat |
| Method | IS | IS | IS | IS | IS |
| StyleGAN2 [9] | 7.29±0.33 | 2.37±0.08 | **1.03**±0.01 | **1.67**±0.05 | 1.31±0.02 |
| ADA [6] | 8.13±0.30 | 2.43±0.06 | 1.01±0.00 | 1.38±0.03 | 1.10±0.01 |
| APA [5] | 7.35±0.27 | 2.37±0.08 | 1.02±0.00 | 1.45±0.01 | **1.43**±0.02 |
| DiffAug [19] | 8.22±0.31 | 2.03±0.05 | 1.01±0.00 | 1.29±0.02 | 1.29±0.01 |
| FastGAN [12] | 7.60±0.30 | 2.28±0.06 | 1.00±0.00 | 1.32±0.02 | 1.33±0.02 |
| FreGAN (Ours) | **8.75**±0.33 | **2.47**±0.06 | 1.00±0.00 | 1.50±0.02 | 1.35±0.01 |

Table 13: The IS (higher is better) scores of our method compared to state-of-the-art methods on **256 × 256 datasets with limited data amounts**.

| | Medici (100 imgs) | Temple (100 imgs) | Bridge (100 imgs) | Wuzhen (100 imgs) |
|---|---|---|---|---|
| Method | IS | IS | IS | IS |
| StyleGAN2 [9] | 1.32±0.03 | **2.04**±0.06 | 1.60±0.03 | 1.93±0.04 |
| ADA [6] | **2.18**±0.04 | 1.98±0.04 | **1.92**±0.03 | 2.02±0.03 |
| APA [5] | 1.24±0.01 | 1.60±0.02 | 1.08±0.01 | **2.43**±0.07 |
| DiffAug [19] | 1.78±0.03 | 1.76±0.02 | 1.68±0.03 | 1.93±0.05 |
| FastGAN [12] | 1.76±0.04 | 1.63±0.02 | 1.55±0.02 | 1.99±0.04 |
| FreGAN (Ours) | 1.69±0.02 | 1.62±0.01 | 1.59±0.03 | 2.12±0.05 |

Table 14: The IS (higher is better) scores of our method compared to state-of-the-art methods on **512 × 512 datasets with limited data amounts**.

| | AnimeFace 120 imgs | ArtPainting 1000 imgs | Moongate 136 imgs | Flat 36 imgs | Fauvism 124 imgs |
|---|---|---|---|---|---|
| Method | IS | IS | IS | IS | IS |
| StyleGAN2 [9] | 1.37±0.02 | 2.79±0.08 | 3.82±0.06 | 2.21±0.04 | 2.32±0.04 |
| ADA [6] | 1.93±0.04 | 3.64±0.12 | 4.12±0.19 | 3.77±0.09 | 2.93±0.07 |
| APA [5] | 1.85±0.06 | **4.65**±0.17 | **4.19**±0.17 | 3.02±0.05 | **3.68**±0.10 |
| DiffAug [19] | 1.19±0.01 | 3.39±0.06 | 2.66±0.07 | 1.48±0.02 | 3.14±0.10 |
| FastGAN [12] | 2.06±0.04 | 4.25±0.12 | 2.96±0.09 | 4.55±0.14 | 3.24±0.07 |
| FreGAN (Ours) | **2.08**±0.03 | 4.17±0.14 | 3.39±0.07 | **5.18**±0.21 | 3.20±0.10 |

Table 15: The IS (higher is better) scores of our method compared to state-of-the-art methods on **1024 × 1024 datasets with limited data amounts**.

| | Shells 64 imgs | Skulls 97 imgs | Pokemon 833 imgs | BrecaHAD 162 imgs | MetFace 1336 imgs |
|---|---|---|---|---|---|
| Method | IS | IS | IS | IS | IS |
| StyleGAN2 [9] | 3.24±0.10 | **3.97**±0.09 | 2.13±0.04 | 1.65±0.03 | 1.92±0.04 |
| ADA [6] | 3.78±0.09 | 2.61±0.08 | 1.41±0.02 | 2.82±0.10 | 3.38±0.09 |
| APA [5] | **3.83**±0.14 | 2.82±0.09 | 1.99±0.44 | **3.01**±0.13 | **3.51**±0.12 |
| DiffAug [19] | 3.14±0.10 | 2.67±0.08 | 2.60±0.07 | 2.72±0.10 | 3.29±0.10 |
| FastGAN [12] | 2.52±0.09 | 2.24±0.08 | **2.63**±0.09 | 2.83±0.04 | 3.07±0.08 |
| FreGAN (Ours) | 2.72±0.06 | 2.47±0.06 | 2.38±0.03 | **3.01**±0.05 | 3.06±0.08 |

## B    Results of Large Datasets

We evaluate the effectiveness of our method on large-scale datasets in Tab. 21 and Tab. 22. Specifically, we use the whole datasets of celebA [13] as training data, and we randomly select 30k images from

Table 16: The IS (higher is better) scores of our method compared to the baseline FastGAN [12] on **AFHQ (∼5k) [1] datasets with more data**.

| Method | AFHQ-Cat (5153 imgs) IS | AFHQ-Dog (4739 imgs) IS | AFHQ-Wild (4738 imgs) IS |
|---|---|---|---|
| FastGAN [12] | 1.93±0.02 | 8.44±0.27 | 5.08±0.08 |
| +Ours | **2.07**±0.05 | **9.23**±0.31 | **5.14**±0.10 |

Table 17: The LPIPS (higher is better) scores of our method compared to state-of-the-art methods on 256 × 256 datasets with limited data amounts.

| | Animal Face | | 100-shot | | |
|---|---|---|---|---|---|
| | Dog (389 imgs) | Cat (160 imgs) | Panda | Obama | Grumpy_cat |
| Method | LPIPS | LPIPS | LPIPS | LPIPS | LPIPS |
| StyleGAN2 [9] | 0.6550±0.0011 | 0.5774±0.0010 | 0.4935±0.0013 | 0.5264±0.0013 | 0.4632±0.0013 |
| ADA [6] | 0.6499±0.0011 | 0.6145±0.0014 | 0.4996±0.0012 | 0.4717±0.0015 | 0.4721±0.0017 |
| APA [5] | 0.6296±0.0013 | 0.6310±0.0013 | 0.5037±0.0011 | 0.4922±0.0012 | 0.4519±0.0011 |
| DiffAug [19] | 0.6301±0.0009 | 0.5453±0.0013 | 0.5144±0.0010 | 0.4727±0.0013 | 0.4426±0.0013 |
| FastGAN [12] | 0.6751±0.0011 | 0.6452±0.0014 | 0.6073±0.0009 | **0.6081**±0.0012 | 0.6077±0.0009 |
| FreGAN (Ours) | **0.6848**±0.0011 | **0.6671**±0.0014 | **0.6089**±0.0010 | 0.6025±0.0011 | **0.6149**±0.0009 |

Table 18: The LPIPS (higher is better) scores of our method compared to state-of-the-art methods on 256 × 256 datasets with limited data amounts.

| Method | Medici (100 imgs) LPIPS | Temple (100 imgs) LPIPS | Bridge (100 imgs) LPIPS | Wuzhen (100 imgs) LPIPS |
|---|---|---|---|---|
| StyleGAN2 [9] | 0.5410±0.0013 | 0.5280±0.0017 | 0.4651±0.0014 | **0.6736**±0.0012 |
| ADA [6] | 0.5182±0.0030 | 0.4389±0.0024 | 0.5341±0.0021 | 0.6086±0.0015 |
| APA [5] | 0.3550±0.0027 | 0.4909±0.0017 | 0.5872±0.0013 | 0.6622±0.0014 |
| DiffAug [19] | 0.4683±0.0017 | 0.5250±0.0014 | 0.5674±0.0011 | 0.6724±0.0013 |
| FastGAN [12] | **0.5298**±0.0024 | 0.5229±0.0015 | **0.5958**±0.0010 | 0.6630±0.0011 |
| FreGAN (Ours) | **0.5298**±0.0024 | **0.5275**±0.0015 | 0.5947±0.0009 | 0.6656±0.0011 |

Table 19: The LPIPS (higher is better) scores of our method compared to state-of-the-art methods on 512 × 512 datasets with limited data amounts.

| | AnimeFace 120 imgs | ArtPainting 1000 imgs | Moongate 136 imgs | Flat 36 imgs | Fauvism 124 imgs |
|---|---|---|---|---|---|
| Method | LPIPS | LPIPS | LPIPS | LPIPS | LPIPS |
| StyleGAN2 [9] | 0.4253±0.0020 | 0.7244±0.0009 | 0.7047±0.0026 | 0.6223±0.0026 | 0.6344±0.0009 |
| ADA [6] | 0.5611±0.0015 | 0.8102±0.0015 | 0.6418±0.0015 | 0.7288±0.0017 | 0.6509±0.0014 |
| APA [5] | 0.5491±0.0017 | 0.8062±0.0014 | **0.7235**±0.0016 | 0.6317±0.0022 | 0.6848±0.0014 |
| DiffAug [19] | 0.4926±0.0005 | 0.7717±0.0016 | 0.5880±0.0015 | 0.4403±0.0005 | 0.6117±0.0023 |
| FastGAN [12] | 0.6188±0.0011 | 0.8344±0.0015 | 0.6603±0.0010 | 0.7939±0.0016 | **0.7028**±0.0010 |
| FreGAN (Ours) | **0.6191**±0.0010 | **0.8439**±0.0016 | 0.6673±0.0016 | **0.7952**±0.0011 | **0.7028**±0.0010 |

Table 20: The LPIPS (higher is better) scores of our method compared to state-of-the-art methods on 1024 × 1024 datasets with limited data amounts.

| | Shells 64 imgs | Skulls 97 imgs | Pokemon 833 imgs | BrecaHAD 162 imgs | MetFace 1336 imgs |
|---|---|---|---|---|---|
| Method | LPIPS | LPIPS | LPIPS | LPIPS | LPIPS |
| StyleGAN2 [9] | **0.5486**±0.0018 | **0.6565**±0.0024 | 0.5870±0.0006 | 0.4604±0.0016 | 0.5321±0.0016 |
| ADA [6] | 0.5268±0.0015 | 0.5857±0.0025 | 0.4050±0.0015 | 0.5524±0.0016 | 0.6648±0.0013 |
| APA [5] | 0.5337±0.0017 | 0.6208±0.0025 | 0.4241±0.0013 | 0.4874±0.0016 | **0.6947**±0.0014 |
| DiffAug [19] | 0.4935±0.0015 | 0.6027±0.0028 | 0.4811±0.0011 | 0.4961±0.0024 | 0.6579±0.0013 |
| FastGAN [12] | 0.4908±0.0012 | 0.6115±0.0028 | 0.5705±0.0008 | **0.5597**±0.0012 | 0.6698±0.0013 |
| FreGAN (Ours) | 0.4957±0.0014 | 0.6112±0.0026 | **0.5770**±0.0008 | 0.5582±0.0011 | 0.6706±0.0012 |

FFHQ [8] as training data. We generate 50k images for quantitative evaluation, we can observe from Tab. 21 and Tab. 22 that our FreGAN also improves the generation quality on large-scale training data.

Table 21: Comparison results of our method compared to the baseline FastGAN [12] on **CelebA dataset with 30k training images**.

| Method | FID ($\downarrow$) | KID ($\downarrow$) | IS | Precision | Recall | Density | Coverage |
|---|---|---|---|---|---|---|---|
| FastGAN [12] | 23.35 | 19.04 | 2.57$\pm$0.07 | 0.69 | 0.25 | 0.60 | 0.35 |
| FreGAN (Ours) | **20.65** | **15.73** | **2.58**$\pm$0.06 | **0.70** | **0.29** | **0.67** | **0.39** |

Table 22: Comparison results of our method compared to the baseline FastGAN [12] on **FFHQ dataset with 30k training images**.

| Method | FID ($\downarrow$) | KID ($\downarrow$) | IS | Precision | Recall | Density | Coverage |
|---|---|---|---|---|---|---|---|
| FastGAN [12] | 26.17 | 14.77 | 3.41$\pm$0.15 | 0.67 | 0.34 | 0.61 | 0.42 |
| FreGAN (Ours) | **23.62** | **12.44** | **3.51**$\pm$0.10 | **0.69** | **0.37** | **0.66** | **0.44** |

In the future, we will also combine our proposed method with large-scale GANs like StyleGAN3 [7] and BigGAN [2] for more comprehensive investigation.

## B.1 More Qualitative Results

We provide more qualitative results of FastGAN and our FreGAN in Fig. 1, Fig. 2, Fig. 3, respectively. Our FreGAN is capable of generating more realistic images with more fine details. The high-frequency components of our generated images contain more information, such as the eyes and mouth of the Panda in Fig. 1, the texture of Moongate in Fig. 2.

We give more generated images of our FreGAN in Fig. 4, Fig. 5, and Fig. 6. We can observe from these figures that our FreGAN can produce vivid images with fine details. The photorealistic generated images indicate the effectiveness of our FreGAN in improving the generation quality of GANs under limited data. However, our FreGAN still struggles in generating photorealistic images when given datasets with limited data but various contents, *e.g.*, only dozens of images, and their contents vary widely. As can be seen from Fig. 6, the overall quality of the generated Pokemon images is not satisfactory because the Pokemon dataset contains multiple categories of Pokemon and only a few images for each character, making it hard to produce high-quality Pokemon images.

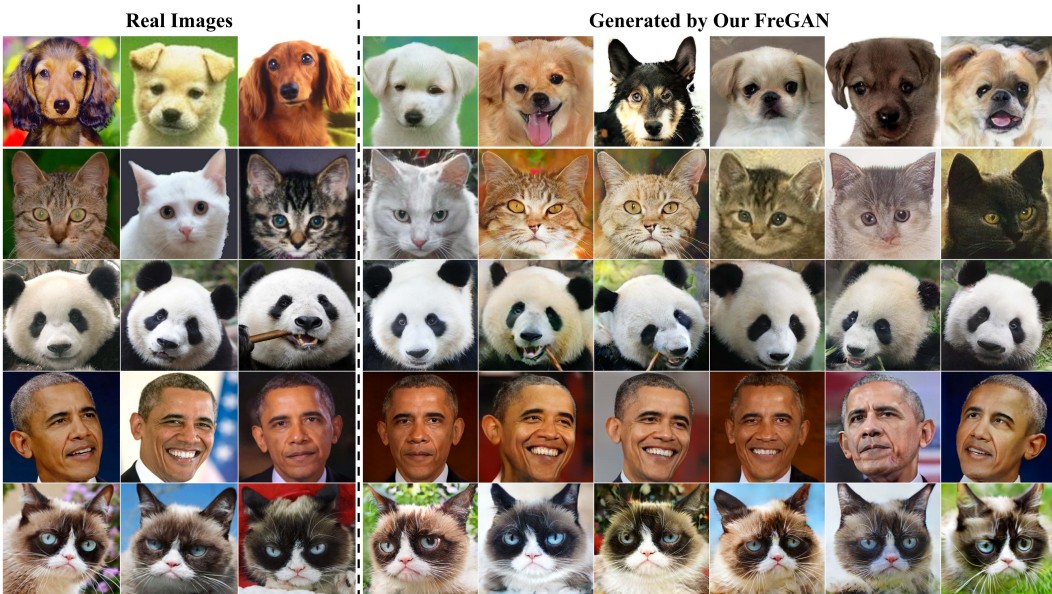

Figure 4: **Qualitative results of our FreGAN.** The left part shows some of real training images and the right part of images are generated by our FreGAN. Our FreGAN is capable of generating photorealistic images with fine details, which is indistinguishable by the discriminator.

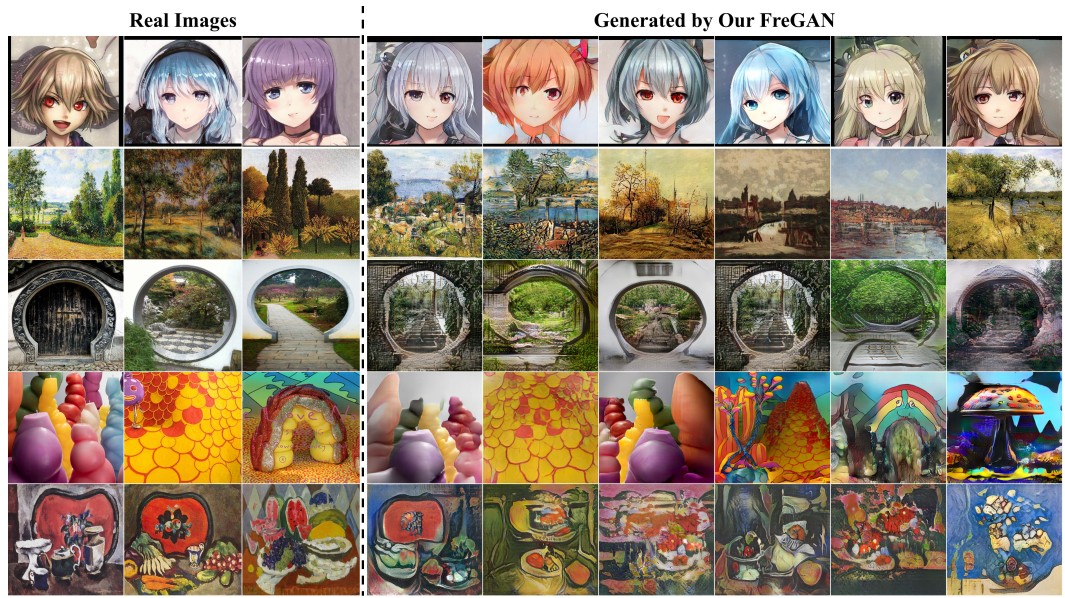

Figure 5: **Qualitative results of our FreGAN.** The left part shows some of real training images and the right part of images are generated by our FreGAN. Our FreGAN is capable of generating photorealistic images with fine details, which is indistinguishable by the discriminator.

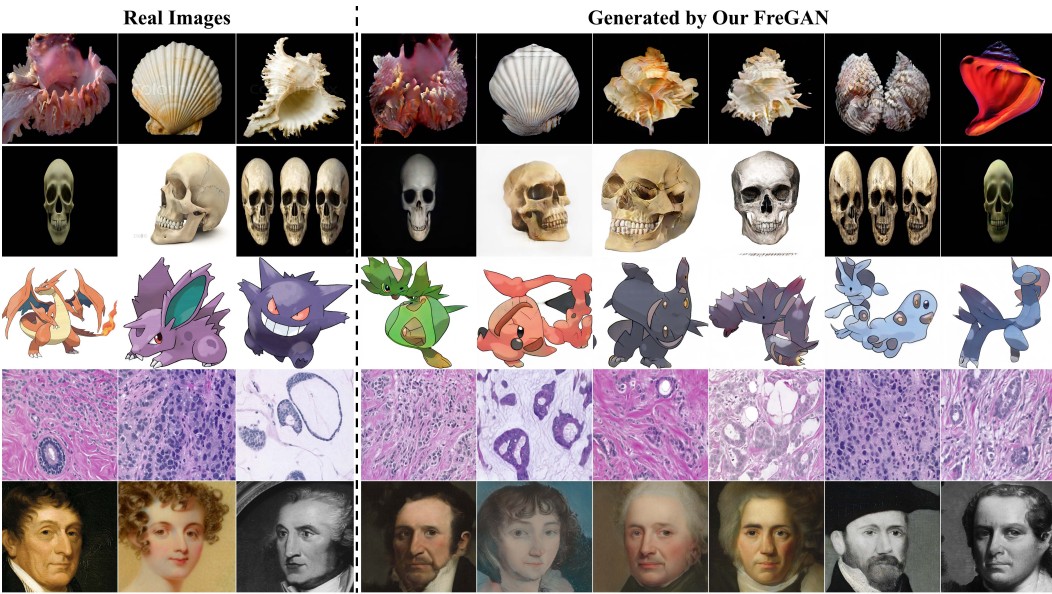

Figure 6: **Qualitative results of our FreGAN.** The left part shows some of real training images and the right part of images are generated by our FreGAN. Our FreGAN is capable of generating photorealistic images with fine details, which is indistinguishable by the discriminator.

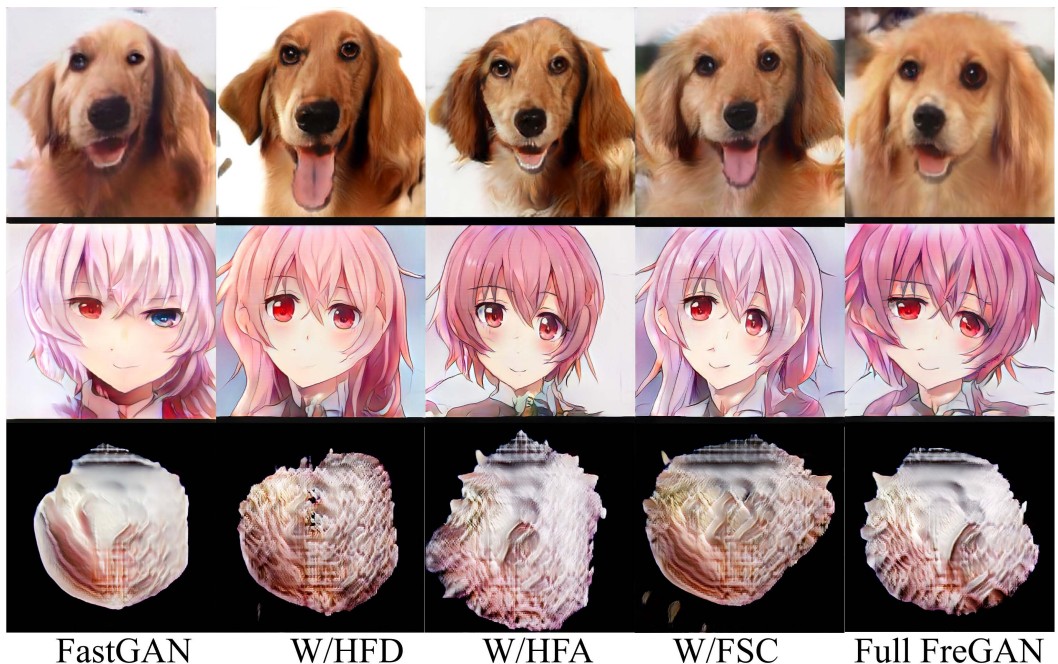

| FastGAN | W/HFD | W/HFA | W/FSC | Full FreGAN |
|---|---|---|---|---|

Figure 7: **Qualitative comparison results of ablation studies.** Each component of our method contributes to the quality of the generated images. HFD improves the fine details like the eyes of the dog and the color of eyes and hairs of the AnimeFace, and HFA makes nostrils, teeth more realistic.

## C   Analysis on generation diversity

We provide the latent space interpolation results of our FreGAN in Fig. 8, from which we can observe that the transition of images generated by different latent codes are smooth and photo-realistic, indicating that our FreGAN promotes the generation quality without compromising the generation diversity. Moreover, we find the closest real images to the generated ones from training data based on LPIPS score, the visualization results are given in Fig. 9, Fig. 10, and Fig. 11. The results demonstrate that our FreGAN learns to produce new images instead of memorizing training images. For example, the body hair color, perspective, and demeanor of dogs are different. The mouth, eyes, hairstyle of AnimeFace are different. And for the Shells dataset in Fig. 11, different generated images that have the closest distance with the same real image are different in color, shape, etc, which further demonstrating that our method improves generation quality meanwhile maintaining diversity. To investigate the training process of our FreGAN, we plot the outputs of the discriminator throughout the training in Fig. 12, the stable curve of our FreGAN demonstrate that our FreGAN can be trained more effective.

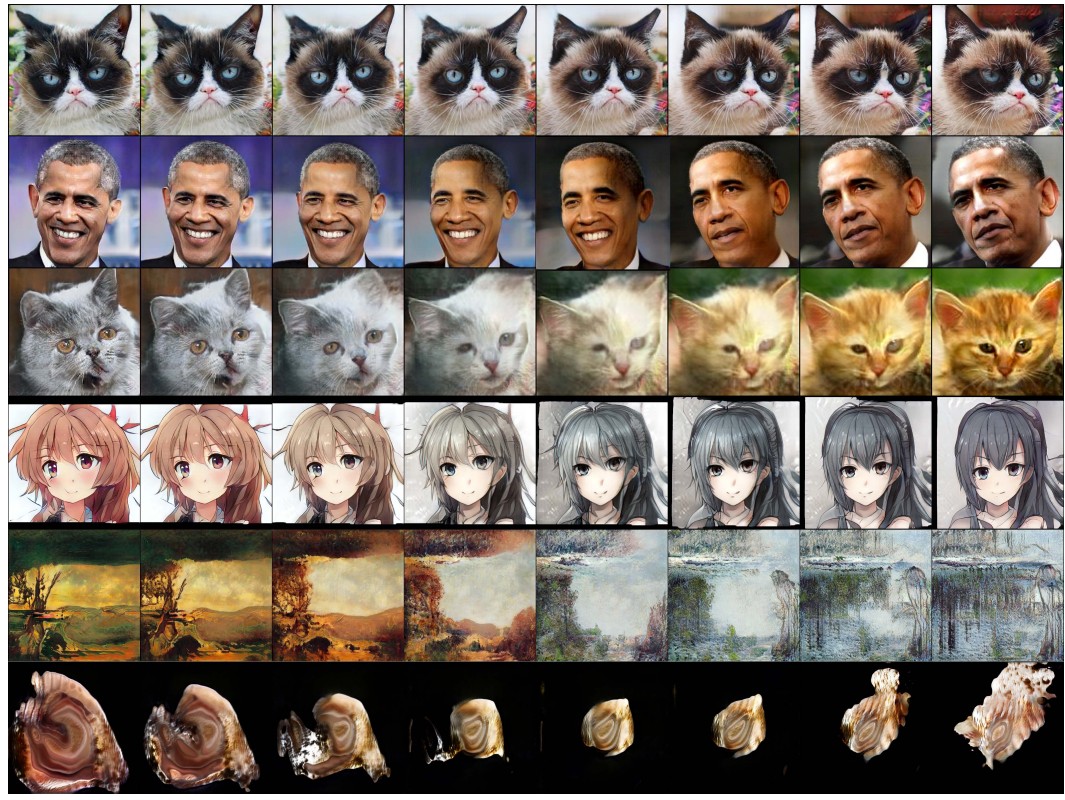

Figure 8: **Latent space interpolation results of our FreGAN.** The smooth transition images suggest that our FreGAN is capable of generating images instead of memorizing images.

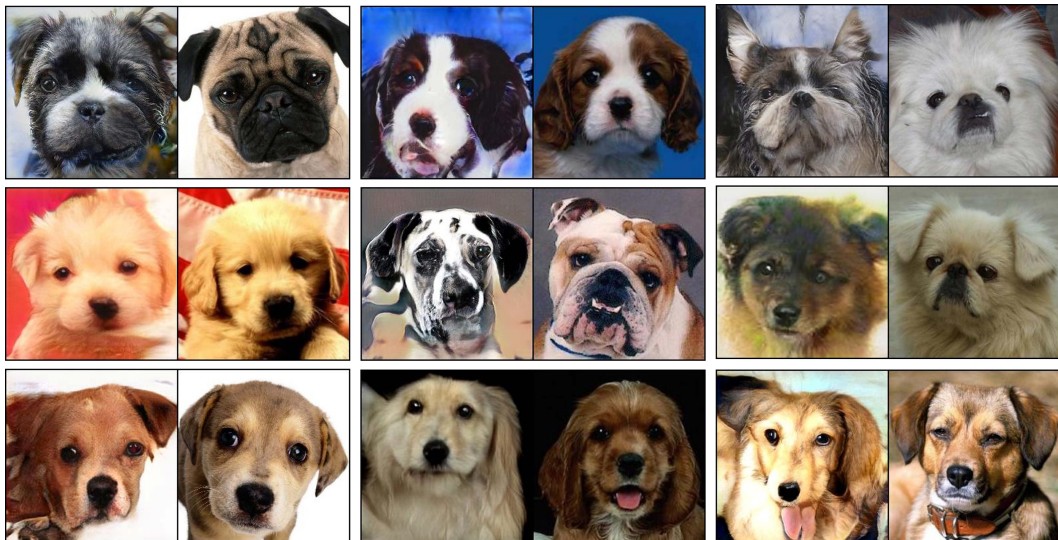

Figure 9: **Nearest real samples to the generated ones on AnimalFace Dog dataset.** For each paired images, the left one is generated by our FreGAN and the right one is the closest image found from the training data. We adopt LPIPS score to measure the similarity of images, *i.e.*, the closest image found from the training data have the largest LPIPS score with the generated one.

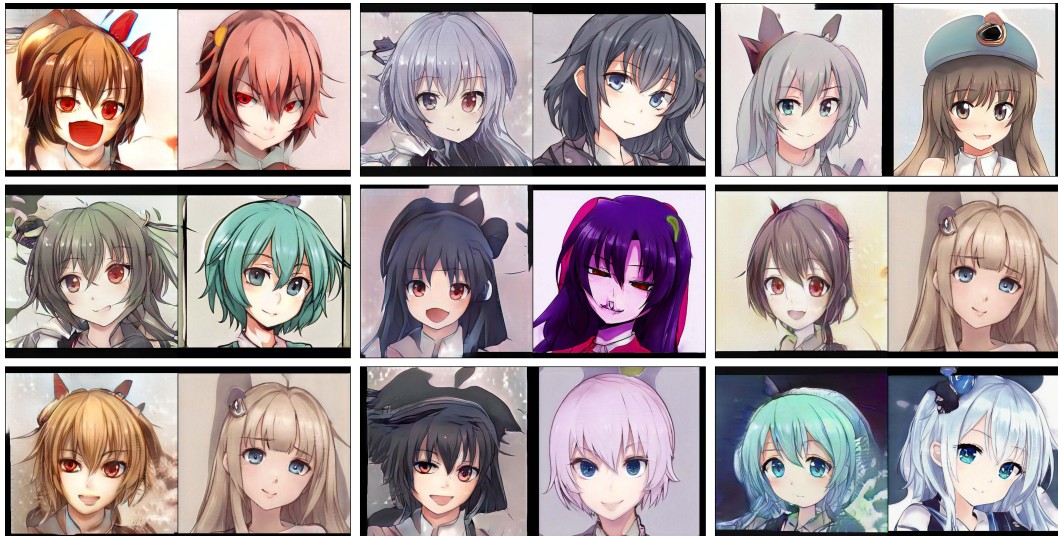

Figure 10: **Nearest real samples to the generated ones on AnimeFace dataset.** For each paired images, the left one is generated by our FreGAN and the right one is the closest image found from the training data. We adopt LPIPS score to measure the similarity of images, *i.e.*, the closest image found from the training data have the largest LPIPS score with the generated one.

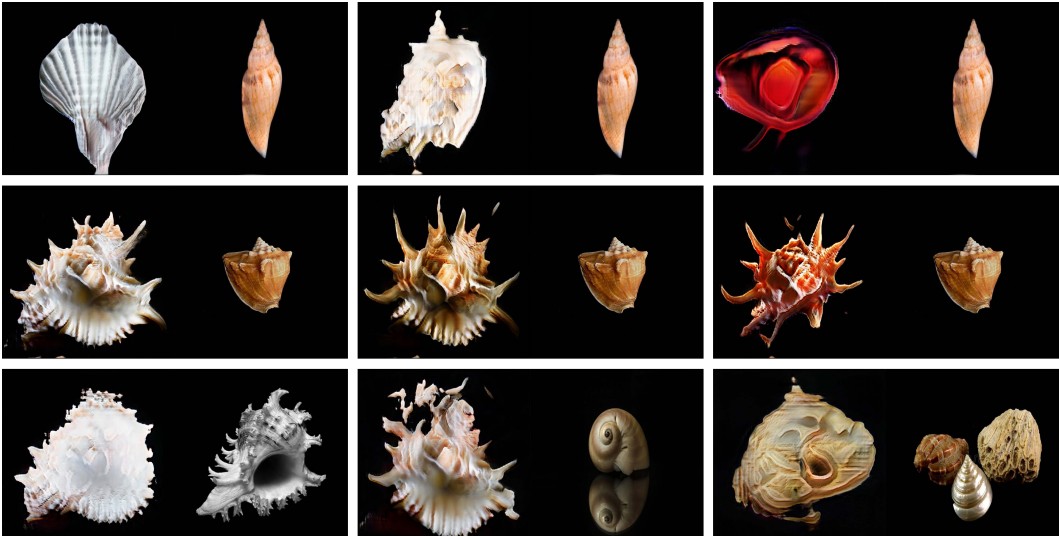

Figure 11: **Nearest real samples to the generated ones on Shells dataset.** For each paired images, the left one is generated by our FreGAN and the right one is the closest image found from the training data. We adopt LPIPS score to measure the similarity of images, *i.e.*, the closest image found from the training data have the largest LPIPS score with the generated one.

## D More analysis on spectral properties of generated images

We first give 2D DWT results of real images in Fig. 13. Then we provide the 2D DWT visualization results in Fig. 14 and Fig. 15, which complement the qualitative comparison results in Figure.4 of the main paper. We can observe that although presented in different ways of visualization, images generated by our FreGAN contain more realistic frequency signals, indicating the efficacy of our proposed techniques. Besides, we compare the averaged 2D power spectrum, one-dimensional slices of the power spectrum, the power spectrum distance, and the statistic (mean and variance power spectrum) in Fig. 16, Fig. 17, Fig. 18, and Fig. 19, respectively. We can observe from these figures that: 1) Our FreGAN can produce more realistic frequency signals compared with other methods; 2) The overlap between the generated images of our FreGAN and the training data is the largest

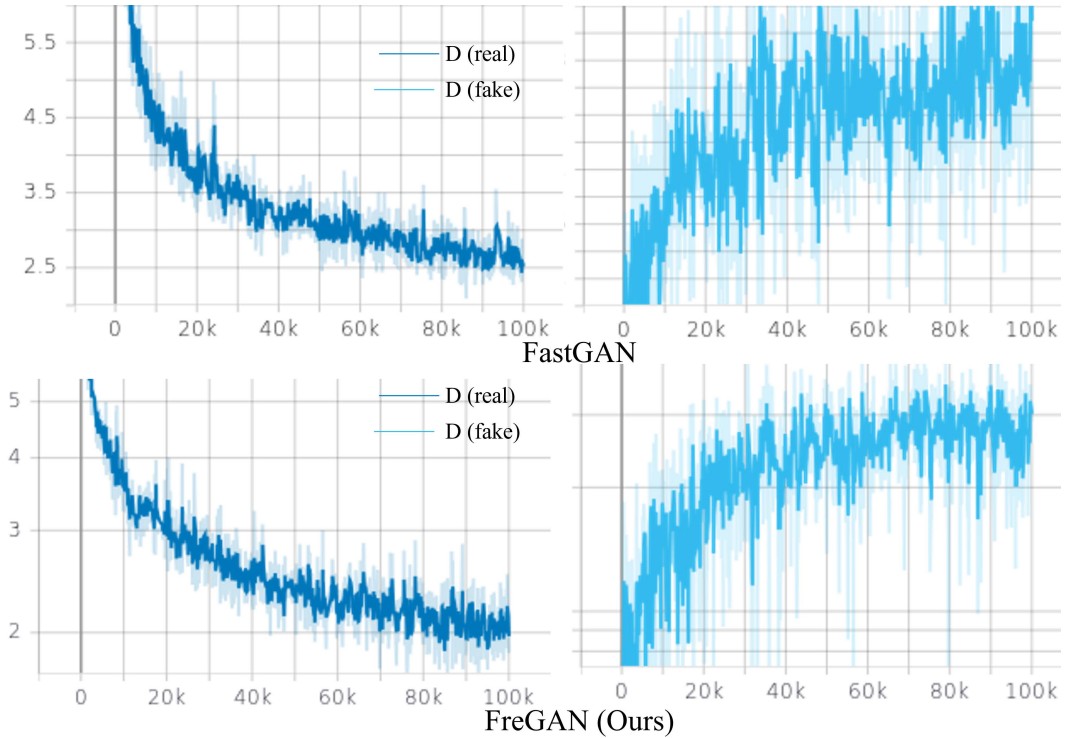

Figure 12: **Comparison results of the outputs of the discriminator throughout the training.** The outputs of the discriminator on fake images of our FreGAN rise more stable and continuous, indicating that our generator is trained more effectively, leading to better generation quality.

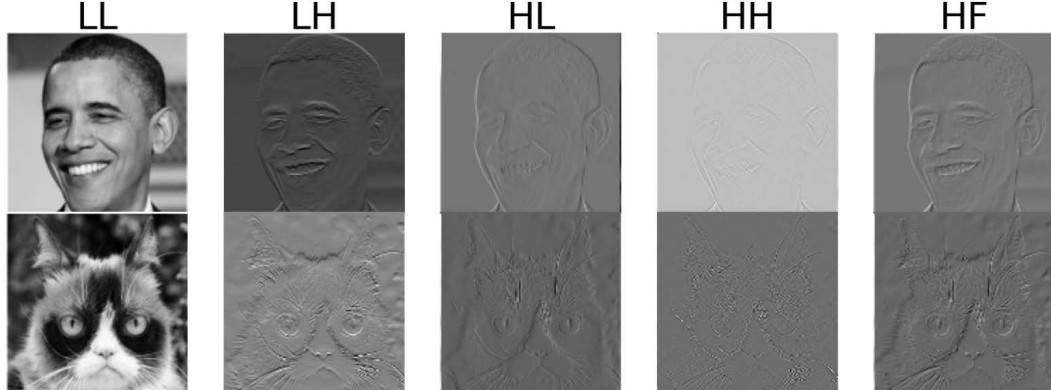

Figure 13: **2D DWT illustration results of differen frequency components.** The low (L) pass filter captures images' overall textures and outlines, and the high (H) pass filter concentrates on details such as the background and edges.

(Fig. 19); 3) Our FreGAN is stable and can consistently produce effective frequency signals (Fig. 17 and Fig. 18);

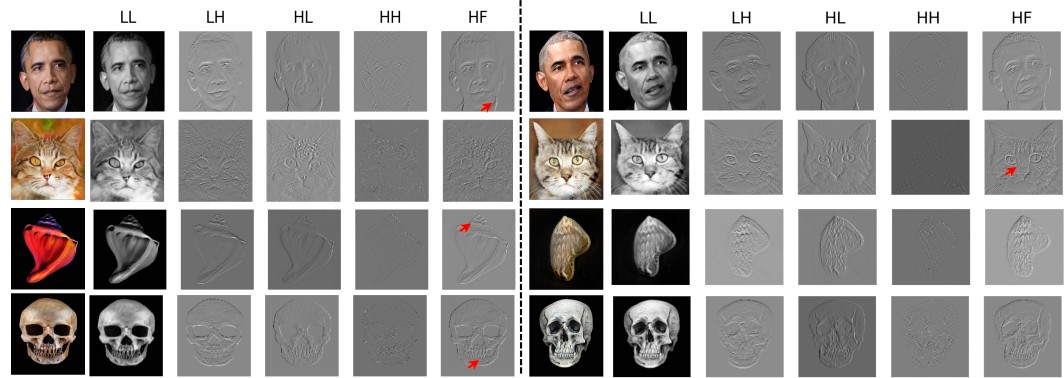

Figure 14: **2D DWT Qualitative comparison results of our FreGAN and the baseline FastGAN.** The images from left to right are generated images, 2D DWT LL, LH, HL, HH, and the combined High-frequency components respectively. Our FreGAN improves the overall quality of generated images and raises the model's frequency awareness, encouraging the generator to produce precise high-frequency signals with fine details.

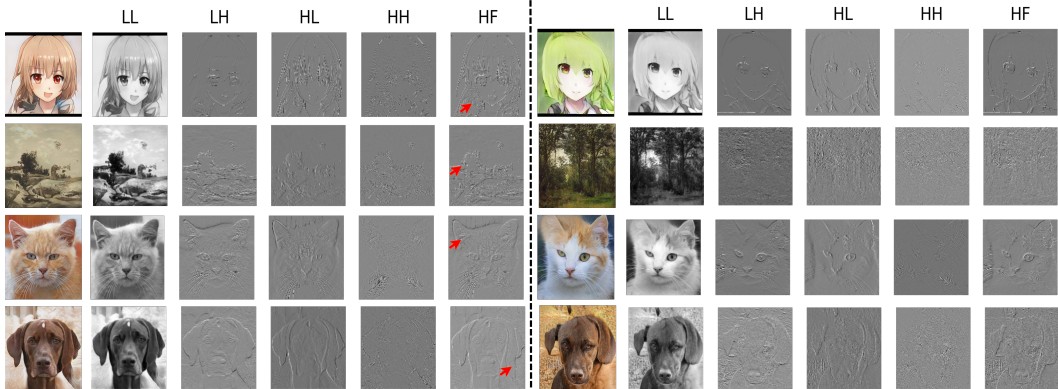

Figure 15: **2D DWT Qualitative comparison results of our FreGAN and the baseline FastGAN.** The images from left to right are generated images, 2D DWT LL, LH, HL, HH, and the combined High-frequency components respectively. Our FreGAN improves the overall quality of generated images and raises the model's frequency awareness, encouraging the generator to produce precise high-frequency signals with fine details.

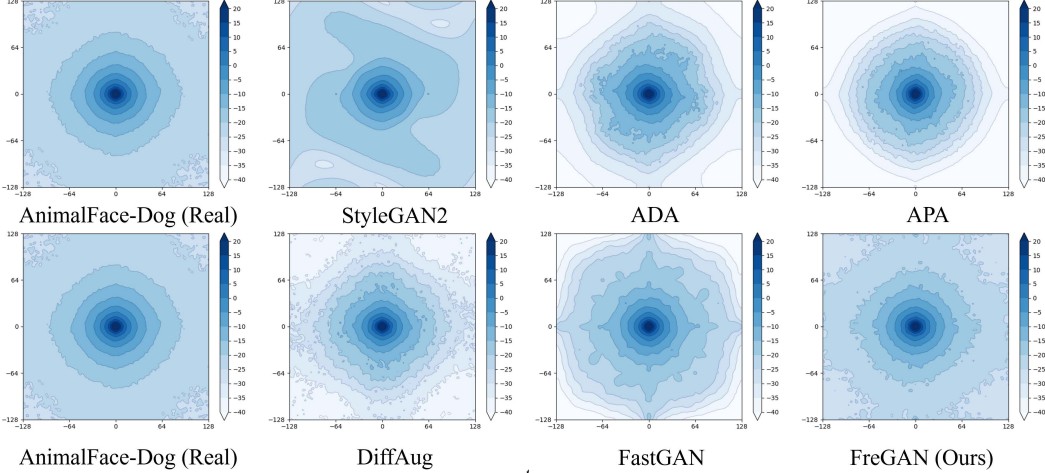

Figure 16: **Comparison results of average 2D power spectrum [7] on Animal Face Dog.** The average 2D power spectrum result for the real data is computed from all training data, and the results of our FreGAN and compared methods are computed from 5k generated images.

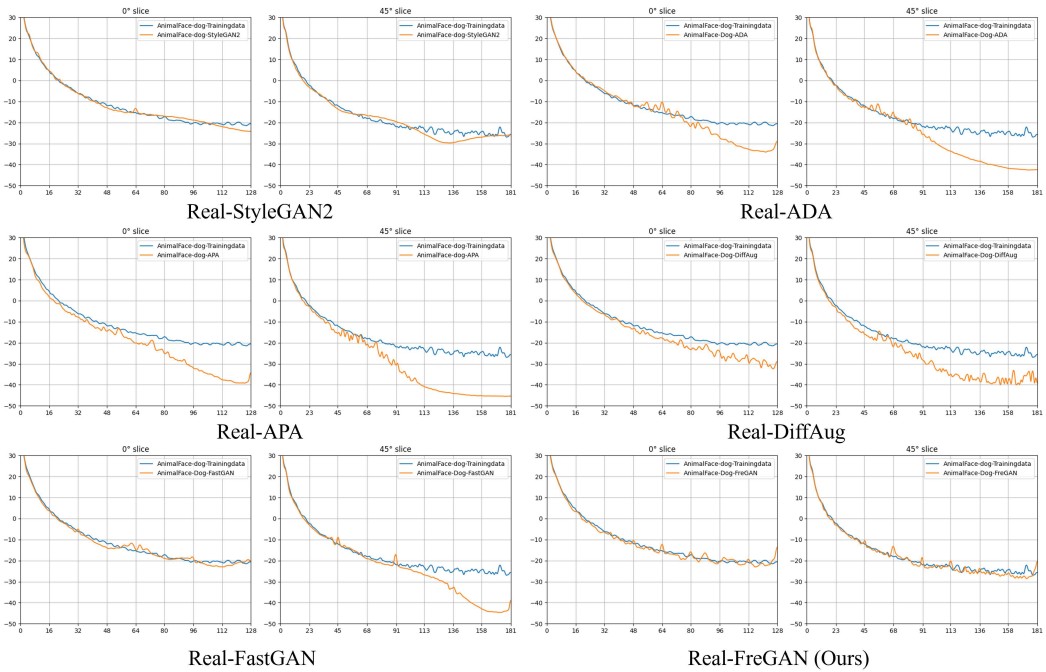

Figure 17: **Comparison results of one-dimensional slices of the power spectrum [7] on Animal Face Dog.** The 1D slices of the spectrum are computed along the horizontal angle (0°) without azimuthal integration [7].