# OpenReview forum: "FreGAN: Exploiting Frequency Components for Training GANs under Limited Data"
_NeurIPS.cc/2022/Conference — NeurIPS 2022 Accept_

### Official Review · Reviewer_BaYJ · 2022-07-03

**Rating:** 5
**Confidence:** 4
**Soundness:** 2 fair
**Presentation:** 2 fair
**Contribution:** 2 fair

**Summary:**

The paper proposes to improve the training of GANs with limited data by means of more careful attention to spectral properties of real and fake images. The proposed model, FreGAN, utilizes wavelets to map the generator’s and discriminator’s intermediate features to the frequency domain. These features are used as input for an additional discriminator’s branch (HFD) to raise the awareness of the discriminator to the frequency domain of images, as well as for an additional loss (HFA) which is used to provide supervision in the frequency domain for the generator. FreGAN is evaluated on a range of limited-size datasets (64-5000 images) and exhibits better FID and KID scores compared to previous few-shot GAN models.


**Questions:**

[1] My major concern is that the design of FreGAN favors the memorization of the training images more than the FastGAN baseline. Given the limited dataset size, the additional discriminator HFD can memorize all the possible HF for real images, and can thus force the generated images to have exactly same HF, which forces to fully reproduce the original training images, and not to generate any new images (as they would have different HF). On the other hand, HFA is a rather restrictive L1 reconstruction loss that also makes the generated images look closer to the exact training examples across all G layers. Missing evaluation of diversity, as well as some visual examples (e.g., 2 memorized examples for “shells” in figure 6 in supplementary) second my concerns. Please comment.

[2] Please also comment on the other concerns 2,3,4.


**Limitations:**

The authors have adequately addressed the limitations and potential negative societal impact of their work.

**Strengths And Weaknesses:**

--- Strengths ---

[1 - Originality] The paper combines the task of training GANs under limited data and the task of addressing the spectral discrepancy between real and GAN-generated images. The experiments demonstrate that raising the frequency awareness helps to achieve both improved image quality (FID, KID) and better rendering of high frequencies (visually). The paper is the first to explore this synergy for this specific task.

[2 - Results] The results in tables look strong. The gain in FID and KID in comparison to the baseline FastGAN is clearly visible and is consistent across different datasets, image resolutions, and the number of training images.


--- Weaknesses ---

[1 - Evaluation (major concern)] In few-shot image synthesis, image quality metrics (FID, KID) can be often improved at the cost of the synthesis diversity [1] (figure 4). For example, the model that perfectly reproduces the training set can achieve near-to-zero FID and KID. The diversity aspect is not evaluated in this work at all. This makes the whole evaluation of the paper unconvincing. Does FreGAN improve FID of FastGAN at the cost of diversity? Is FreGAN memorizing training images more than FastGAN? Are the interpolations in the latent space of FreGAN smooth? It is not possible to answer with the provided figures and tables.

In addition to FID or KID, I suggest adding a metric that explicitly measures the diversity of generated images, as done in some previous works. One can choose pairwise LPIPS as in [2] (tables G,H), or intra-cluster LPIPS as in [3] (table 2). Moreover, it would be good to measure the average LPIPS from generated images to the nearest example from the training set, which would help to assess the degree of memorization of the training set ([4], table 8). Lack of diversity is also partially supported by low recall across different datasets reported in the supplementary materials. On the qualitative side, it would be good to add examples of latent space interpolations as in [4] (figure 5) and the nearest real neighbors to some of generated images ([4], figure 12-13). Overall, without this analysis, given the size of the used datasets, only improving FID and KID is not a convincing result to me.


[2 - Claim (major concern)] The paper claims that the proposed method alleviates the unhealthy competition between G and D (lines 11, 47, 54) in the low data regime. The caption of figure 5 says that the FreGAN generator can better deceive the discriminator. The justification for the claim is claimed in the loss curves plot in figure 5(a).

In fact, what is seen in figure 5(a) seems to say the opposite. The discriminator loss of FreGAN is lower than the one of FastGAN (even though it has an additional term L^HF). This indicates that the discriminator can dissect real and fake images more confidently. It would be interesting to see the curves of the discriminator outputs during training (e.g., as in [5], figure 1(b-c)), where this effect should be more visible. Consequently, based on the plot 5(a), the FreGAN generator struggles to fool the discriminator more than FastGAN. This is confirmed by the larger G loss for FreGAN. It is thus not clear how the better GAN equilibrium is quantified by the authors, so one of the main claims is not supported.



[3.1 – Clarity (major concern)] Generally, it is not clear why the technical proposals of the paper are introduced as techniques for training GANs in low data regimes. The frequency bias for GANs exists not only when the training data size is limited [6]. In principle, any GAN model should benefit from generating images with better spectral properties (e.g., the motivation in lines 33-39 also fits to training GANs on large datasets). The explanation on mitigating unhealthy G and D competition is, in my opinion, unsupported (see Weaknesses-2).

Why are the introduced modifications tested only in low-data regimes? Do the proposed techniques help to improve the quality of images or spectral properties of GANs trained on large datasets?


[3.2 – Related work (major concern)] The proposed method is by design aimed to improve the quality of images in the frequency domain. There are other works with similar motivation (e.g., [6] as an overview). These methods can be potentially placed instead of the techniques introduced in section 3.2-3.4. It is not clear whether the effect of the proposed techniques could not be achieved with already existing techniques mitigating spectral biases of GANs. A proper comparison to previous works is expected.

[3.3 – Evaluation (major concern)] The proposed method is aimed to improve the quality of images in the frequency domain, but this aspect is not evaluated quantitatively. The only provided comparisons of high/low frequencies are visual, which makes it subjective, especially since ground truth spectrums to compare to are not available. Quantitative assessment of GAN images spectrums was approached before, for example by measuring the accuracy of a binary classifier trained on spectrum features of real and fake images [7]. A proper quantitative comparison of spectral properties of images for different models is expected.

[4 – Clarity (moderate concern)] The motivation of the frequency alignment loss (lines 151-161) is not clear. Firstly, what is meant by “G can only synthesize arbitrary frequency signals”? The synthesis of G is guided with the discriminator loss, which provides supervision on the image realism, which includes the intensity of details of different frequencies. So they should not be arbitrary. Secondly, why is HDA called a “regularizer for D”, while it is included only to the objective of G?

Finally, eq. (4) is a form of a reconstruction loss, but it is computed between D’s features of a real image and G’s features of a randomly sampled fake image. These images are not necessarily aligned spatially, so it is not clear why imposing a pixel-wise L1 reconstruction loss is meaningful. Please explain.


--- References ---

[1] Few-Shot Adaptation of Generative Adversarial Networks. Robb et al. Arxiv, 2020.

[2] Generating Novel Scene Compositions from Single Images and Videos. Sushko et al. Arxiv, 2021.

[3] Few-shot Image Generation via Cross-domain Correspondence. Ojha et al. CVPR 2021.

[4] Towards Faster and Stabilized GAN Training for High-fidelity Few-shot Image Synthesis. Liu et al. ICLR 2021.

[5] Training Generative Adversarial Networks with Limited Data. Karras et al. NeurIPS 2020.

[6] On the Frequency Bias of Generative Models. Schwarz et al. NeurIPS 2021.

[7] Watch your Up-Convolution: CNN Based Generative Deep Neural Networks are
Failing to Reproduce Spectral Distributions. Durall et al. CVPR 2020.

---

> ### Author Response · Authors · 2022-08-02
> **Response to Reviewer BaYJ Part-1**
>
> Thank you for the review. We highlighted the revised content in the revised version, please refer to. The Answers to specific questions:
>
> Q1: The diversity aspect is not evaluated. Adding a metric to measures the diversity of generated images and qualitative analysis on the latent space interpolation and nearest real neighbors.
>
> A1: Thanks. As suggested, we add LPIPS metric to explicitly measure the diversity of generated images. Specifically, we compute the LPIPS of all the paired images of 5k images generated by each method and report the average and standard deviation of LPIPS scores. The results are presented in Table 17 (page 9), Table 18 (page 9), Table 19 (page 9), and Table 20 (page 9) in the appendix. Our FreGAN achieves either the best or comparable results on the LPIPS scores, demonstrating that our FreGAN improves the generation quality without remembering the training images more than FastGAN. Figure 8 (page 13 of the appendix) shows the latent space interpolation of our FreGAN, and we can see that the transition of images generated by different latent codes is smooth and photo-realistic, indicating that our FreGAN promotes generation quality without compromising generation diversity.
>
> Furthermore, we find the closest real images to the generated ones from training data based on the LPIPS score [4], and the visualization results are given in Figure 9 (page 13), Figure 10 (page 14), and Figure 11 (page 14) of the appendix. The results demonstrate that our FreGAN learns to synthesize new images rather than memorize training images. For example, the body hair color, perspective, and demeanor of dogs are different in Figure 9 (page 13). The mouth, eyes, hairstyle of AnimeFace are all unique in Figure 10 (page 14). Notably, for the Shells dataset in Figure 11 (page 14), different generated images that have the closest distance with the same real image are different in color, shape, and so on, demonstrating that our method improves generation quality while maintaining diversity.
>
> Q2: The claim about alleviating the unhealthy competition between G and D is not supported.
>
> A2: Thanks. The competition between G and D is unbalanced because D can see both real and fake data, whereas G can only learn real data from D's feedback. Our HFA module encourages G to generate realistic high-frequency signals by providing he high-frequency information of the real data to G as a self-supervised signal. Figure 5 (a) (page 9) of the main paper shows that our generator achieves higher scores during training, indicating that it can produce more realistic images and thus deceive the discriminator better. Our FreGAN's discriminator loss is higher than FastGAN's early in training but lower later in training, indicating that our D can be better trained and thus provide meaningful guidance to G. Furthermore, the discriminator scores of FastGAN are always higher than the generation scores, resulting in disequilibrium, whereas our FreGAN reduces the gap toward disequilibrium, improving synthesis quality. Furthermore, as suggested, we provide the comparison results of the outputs of the discriminator throughout the training [5] in Figure 12 (page 15) of the appendix. The outputs of the discriminator on fake images of our FreGAN rise more stable and continuous, indicating that our G is being trained more effectively, resulting in higher generation quality.
>
> Q3-1: Why only tested in low-data regime? Does the proposed techniques improve the generation quality on large datasets?
>
> A3-1: Thanks. For the following reasons, we only tested our method on low-data scenarios: 1) Image generation in low-data regime is more challenging, and it is of great application and research significance to improve generation quality in scenarios with limited data; 2) When data is limited, the problem of the model struggling to fit high-frequency signals is magnified, so improving the model's frequency awareness in low-data regime is critical; 3) Due to limited computing resources, we cannot afford training GANs on large-scale datasets, for example, we need about 10 days to train StyleGAN2 for 25000 kimg on the FFHQ dataset with 2 V100 GPUs.
> Our FreGAN aims at improving the model’s frequency awareness and is not constrained by the scale of training data.
>
> We evaluate the effectiveness of our method on large-scale datasets, i.e., CelebA and FFHQ in Table 21 (page 10) and Table 22 (page 10) of the appendix. Specifically, we use the entire CelebA datasets as training data, and we randomly select 30k images from FFHQ as training data. We train our FreGAN and FastGAN for 200k iterations and generate 50k images for quantitative evaluation. Table 21 (page 10) and Table 22 (page 10) show that our FreGAN also improves the generation quality on large-scale training data over FastGAN. We will also combine our proposed method with large-scale GANs such as StyleGAN3 and BigGAN for a more comprehensive investigation.

---

> > ### Author Response · Authors · 2022-08-02
> > **Response to Reviewer BaYJ Part-2**
> >
> > Q3-2: Compare with previous works (e.g., [6] as a overview) to show the difference and irreplaceability of the proposed method.
> >
> > A3-2: Thanks. [6] investigated the frequency bias of the generator and the discriminator and unified the efforts made to mitigate the spectral discrepancies. [6] also found that the design of the discriminator plays an important role in training GANs. Our FreGAN, which aims to fully utilize the frequency information of limited data, improves the model’s frequency awareness from both the generator and discriminator perspectives. Specifically, we perform a high-frequency discriminator (HFD) which concentrates on learning the frequency information of input images by directly decomposing intermediate features of the discriminator. We propose high-frequency alignment (HFA) to involve the frequency information from D as a self-supervised signal to encourage the generator to produce more plausible frequency signals, which also mitigates the unbalanced competition between G and D. We further introduce frequency skip connection (FSC) in the generator to prevent the loss of high-frequency information. Our proposed techniques complement each other and improve the generation quality, and none of existing methods mitigate the unbalanced competition between G and D from frequency domain perspective. Our FreGAN also differs from SWAGAN in four ways as shown in $\textbf{A1 to Reviewer Xdtv}$, we will review more related works and analysis the differences of our FreGAN with existing methods.
> >
> > Q3-3: A proper quantitative comparison of spectral properties of images for different models is expected.
> >
> > A3-3: Thanks. As suggested, we compare the 2D power spectrum, one dimensional slices of the power spectrum, the power spectrum distance, and the spectral statistic [7] (mean and variance of power spectrum) in Figure 16 (page 16), Figure 17 (page 17), Figure 18 (page 18), and Figure 19 (page 19) in the appendix. These results consistently indicate that our FreGAN performs better in producing frequency signals, facilitating the quality of the generated images.
> >
> > Q4-1: The motivation of frequency alignment loss is not clear. What is meant by “G can only synthesize arbitrary frequency signals?”
> >
> > A4-1: Thanks. The discriminator can indeed provide supervision on producing frequency signals to G, but this is insufficient. This is because D can learn the frequency information from both real and generated data, whereas G can only obtain guidelines from D, making the competition unbalanced. As a result we introduce HFA, which involves the high-frequency information from D as self-supervised signal to regularize G. G is then encouraged to generate realistic high-frequency signals, without our HFA, G cannot grasp the high-frequency information of the real data, which is why we say that G can only synthesize arbitrary frequency signals.
> >
> > Q4-2: Why is HFA called a “regularizer for D”, while it is included only to the objective of G?
> >
> > A4-2: Thanks. We apologize for misleading you, here is a typo. HFA is a regularizer for G that provides guidance of producing high-frequency signals to the generator, thus is included to the objective of G.
> >
> > Q5: Why imposing a pixel-wise L1 reconstruction loss in Eq (4) is meaningful?
> >
> > A5: Thanks. Eq (4) encourages the generator to synthesize realistic frequency signals, guided by the wavelet decomposed high-frequency signals of real images obtained from the discriminator. Because we concentrate on the high-frequency frequency signals here. Therefore, the L1 alignment loss is sufficient to enables G to generate more realistic high-frequency information, and spatial and semantic alignment are unnecessary. Besides, qualitative (Figure 7 on page 12 of the appendix) and quantitative (Table 5 on page 8 of the main paper) analysis of the experimental results demonstrate that HFA can indeed improve the generation quality and contribute to the generation details. Nonetheless, we believe that truncating the loss in Eq (4) may improve G’s to generate high-frequency signals even further, which we will test in the future.
> >
> > Hope that the above discussions address your concerns. Thanks for your effort and constructive suggestions again.

---

> > > ### Author Response · Authors · 2022-08-08
> > > **Response to Reviewer BaYJ**
> > >
> > > Dear Reviewer BaYJ,
> > >
> > > Thank you so much for your effort in reviewing this submission!
> > >
> > >  We have tried our best to address your concerns above the response. Would you mind taking a look and letting us know what you think?
> > >
> > > Please let us know if you have any additional questions, we are happy to clarify them, thanks for your time and effort again.
> > >
> > > Best, Authors.

---

### Official Review · Reviewer_Xdtv · 2022-07-07

**Rating:** 4
**Confidence:** 4
**Soundness:** 3 good
**Presentation:** 2 fair
**Contribution:** 2 fair

**Summary:**

This paper aims to stabilize GAN training in the low-data regime by incorporating wavelet information into the generator and discriminator. Specifically, the paper proposes three measures to improve image synthesis in the low-data regime: i) frequency skip connections (FSC), ii) an additional high-frequency discriminator (HFD), and iii) high-frequency alignment (HFA). The proposed method improves over the baselines on various datasets and for various metrics.


**Questions:**

Please add the comparisons to SWAGAN and ProjectedGAN discussed under weaknesses or justify why they are not applicable.

What is wavelet unpooling? Is it NN upsampling/bilinear upsampling in the wavelet domain? Please define a mathematical operator for this to make it clear.

Fig.3: what is the color scheme? I am confused, shouldn’t the lowpass filter (L) lead to a blurry version of the original image instead of these weird color distortions?

Lastly, adding a spectral analysis of the generated images (see e.g. Schwarz et al) would be instructive to validate the claim that the proposed method ‘raises the frequency awareness’ of the generator.


**Limitations:**

The results in the appendix indicate that recall is very low for all methods and all datasets. While this limitation seems not specific to the proposed method, it should be discussed in more detail in the paper.

**Strengths And Weaknesses:**

Strengths:

Generally, the idea to exploit frequency information for image synthesis in the low data regime is interesting as it might provide additional guidance to the model.
The proposed method is evaluated on a multitude of datasets and metrics. Further, the proposed design choices are validated in an ablation study that demonstrates their effectiveness.

Weaknesses:

The paper is missing any comparison to SWAGAN[1]. While [1] is not aiming at the low-data regime specifically, it features many similarities to the proposed method as it considers image synthesis in the wavelet domain. There needs to be a detailed analysis on how the proposed method differs from SWAGAN as well as to include results for [1] as a baseline. Without this, it is difficult to judge how novel the contributions of this work are.
The paper uses FastGAN as baseline. However, ProjectedGAN [2] improves over FastGAN, also for small datasets. The comparison to ProjectedGAN is missing from the related work and the experimental analysis and thereby lacks comparison to one important state-of-the-art method.
It would also be interesting if the proposed method can be combined with ProjectedGANs for even further improvements.
The language is often colloquial, see misc.

[1] SWAGAN: A Style-based WAvelet-driven Generative Model, Gal et al, SIGGRAPH 21
[2] Projected GANs Converge Faster, Sauer et al, NeurIPS 21

Misc:
L.43, 106 “employ *a* high-frequency discriminator (HFD) and frequency skip connection*s* (FSC)”
L.45, 141, 151, etc  synthesis -> synthesize
L.54,101, etc ‘unhealthy’ -> unbalanced
L.71 However, their (StyleGAN/BigGAN) success is mainly from sufficient training data” - I disagree. The main success of StyleGAN/BigGAN is due to architectural choices, carefully tuned training strategy, and regularization. Nonetheless, it is correct that their performance degrades with limited data.
L.90 prove -> demonstrate
L.112 the Haar transform does not transform an image into the frequency domain, but rather to the wavelet domain.
L.138 add a pointer to ablation where this is shown
L.198 Implement -> Implementation

---

> ### Author Response · Authors · 2022-08-02
> **Response to Reviewer Xdtv**
>
> Q1: Compare with SWAGAN [1] and add a detailed analysis on the differences of the proposed method with SWAGAN.
>
> A1: Thanks. SWAGAN is not a method designed specifically for training GANs with limited data, and none of existing few-shot GANs adopt it as a compared baseline, so we did not compare with SWAGAN at first. Although both SWAGAN and our FreGAN aim to improve the model’s frequency awareness, our FreGAN differs from SWAGAN in three ways: 1) SWAGAN works in the frequency space by predicting the coefficients of basis functions of increasing frequency, whereas the basis functions of our frequency decomposition, determined by given low and high pass filters, do not require predictions and affine transformation layers for the coefficients; 2) SWAGAN progressively generates image decompositions in the frequency domain at the image level, requiring additional up/down sampling to convert images to higher/lower resolution. Our FreGAN is more flexible because FreGAN directly decomposes intermediate features of D and G into the wavelet domain; 3) Instead of using inverse wavelet transformation (IWT) to convert decompositions into spatial domain as in SWAGAN, our high-frequency discriminator (HFD) directly learns the frequency information in the wavelet domain, avoiding high-frequency loss caused by IWT and encouraging the generator to produce plausible high-frequency signals; 4) Our high-frequency alignment (HFA) involves the frequency information from D as a self-supervised signal to guide G, providing better guidance to G and alleviating the unbalanced competition between G and D by showing information of real images to G, which on one has done in the previous works.
>
> We include SWAGAN as the compared method in Table 1 (page 7) and the related work in Section 2 (page 3) of the main paper. We can observe from Table 1 (page 7) that our FreGAN outperforms SWAGAN by a significant margin. Due to time limitation, we only report results of 256x256 datasets, more quantitative results will be added later.
>
> Q2: Compare with ProjectedGAN [2], and it would be interesting to combine our method with ProjectedGAN.
>
> A2: Thanks. Because the datasets we use have less overlap with ProjectedGAN, we did not initially use it as a comparison method. We will reimplement ProjectedGAN on all datasets that we use and compare it to our FreGAN. Because ProjectedGAN improves GANs by projecting generated and real images into a pretrained feature space, it is orthogonal to the method we propose. As a result, we believe that combining our method with Projected GANs will result in further improvements; we will investigate this as suggested.
>
> Q3: Language is colloquial.
>
> A3: Thanks. We reviewed our paper and fixed them carefully, the modified content is highlighted in the revised version.
>
> Q4: What is wavelet unpooling?
>
> A4: Thanks. For wavelet unpooling, we first apply a component-wise transposed-convolution on the signal of each component and then sum all resulted features up to precisely reconstruct the original image from wavelet domain. The PyTorch-like pseudocode of wavelet unpooling is given in Algorithm 1(page 2) of the appendix.
>
> Q5: Color Scheme of Figure3. Low-pass filter should lead to a blurry version of the original image.
>
> A5: Thanks. We directly save the output features of the Haar wavelet for visualization in Figure 3 (page 4 of the main paper), and different colors of LL indicate different activations to the features of images. And the high pass filter concentrates on fine details, thus the activations are not as colorful as low pass filter. We present the 2D DWT visualization results of Figure 3 (page 4) and Figure 4 (page 6) in the main paper in Figure 13 (page 15), Figure 14, and Figure 15 (page 16) in the appendix, which also suggest that low-pass filter captures overall textures and outlines, whereas high-pass filter concentrates on fine details such as vertical and horizontal edges.
>
> Q6: Add spectral analysis of generated images.
>
> A6: Thanks. In the appendix, we compare the 2D power spectrum, one-dimensional slices of the power spectrum, the power spectrum distance, and the statistic (mean and variance power spectrum) in Figure 16 (page 16), Figure 17 (page 17), Figure 18 (page 18), and Figure 19 (page 19). These results show that our FreGAN performs better in producing frequency signals, which facilitates the quality of the generated images.
>
> Limitations: recall is very low for all methods and all datasets.
>
> A: Thanks. We infer that the recall is very low for all methods and all dataset because:1) The diversity of the training data is relatively low, for example, 100-shot-Obama only contains 100 images of Obama, and 2) recall is not suitable for evaluation in scenarios with limited data.
>
> Hope that the above discussions address your concerns. Thanks for your effort and constructive suggestions again.

---

> > ### Comment · Reviewer_Xdtv · 2022-08-03
> > **Response**
> >
> > Dear authors,
> >
> > thank you for addressing my concerns in your rebuttal.
> >
> > You mentioned that initially you did not compare with Projected GANs "because the datasets we use have less overlap with ProjectedGAN". However, similar to your work, ProjectedGAN reports numbers on Pokemon, AFHQ-Cat, AFHQ-Dog, AFHQ-Wild.
> > The reported FID on these datasets in Projected GANs is consistently better than for your method:
> > |              | AHFQ-Cat | AHFQ-Dog | AHFQ-Wild | Pokemon |
> > |--------------|----------|----------|-----------|---------|
> > | ProjectedGAN | 6.62     | 20.75    | 6.37      | 38.88   |
> > | FreGAN       | 2.16     | 4.52     | 2.17      | 33.96   |
> >
> > From your rebuttal it is not clear to me that combining your approach with the ProjectedGAN architecture can indeed improve these results further. Could you please either demonstrate this or show that for smaller datasets ProjectedGAN works worse or has significant disadvantages over your approach?

---

> > > ### Author Response · Authors · 2022-08-03
> > > **Response to Reviewer Xdtv**
> > >
> > > Dear Reviewer Xdtv,
> > >
> > > Thank you a lot for the review and the response.
> > >
> > > We will perform experiments to demonstrate the performance of combining our method with ProjectedGAN.
> > > Experimental results and analysis will be given as soon as possible in the next few days.
> > >
> > > Please let us know if you have any additional questions, thanks for your time and effort again.

---

> > > ### Author Response · Authors · 2022-08-07
> > > **Response to Reviewer Xdtv**
> > >
> > > Dear Reviewer Xdtv,
> > >
> > > Thank you a lot for the review and the response again.
> > >
> > > As suggested, we combine our proposed techniques with ProjectedGAN to investigate the compatibility of our method. We use the official ProjectedGAN codes to implement our proposed techniques while leaving other details unchanged. Specifically, for the discriminator, we first employ Haar wavelet transformation to decompose the projected features of ProjetedGAN into wavelet domain, then we apply additional convolutional operations to the high-frequency components (HFD), improving the discriminator’s frequency awareness of ProjectedGAN. For the generator, we perform Haar wavelet transformation on the intermediate features of the generator of ProjectedGAN and employ our frequency skip connection (FSC) to prevent frequency information loss. The frequency components of the discriminator and generator are then aligned using high frequency alignment (HFA). We train the original ProjetedGAN and the version combined with our method for 1M images on 2 V100 GPUs, with the batchsize set to 16, and x-flips enabled. The FID comparison results are given as follows :
> > > | Datasets    | 100--shot-Panda   | 100--shot-Grumpy_cat | 100--shot-Obama    | AnimalFace-Cat | AnimalFace-Dog |
> > > | ----------- | ------- | ---------- | -------- | ------------- | ------------- |
> > > | ProjetedGAN | 7.34    | 17.75      | 26.34    | 26.50         | 22.51         |
> > > | +Ours       | 7.23    | 17.01      | 25.07    | 26.21         | 21.77         |
> > >
> > > | Datasets    | AFHQ-dog | AFHQ-cat    | AFHQ-wild | Pokemon256    | Pokemon1024   |
> > > | ----------- | ------- | ---------- | -------- | ------------- | ------------- |
> > > | ProjetedGAN | 4.52    | 2.16       | 2.17     | 26.36         | 33.96         |
> > > | +Ours       | 4.52    | 2.13       | 2.09     | 25.31         | 34.07         ||
> > >
> > > ProjectedGAN is a strong baseline that improves synthesize quality significantly by projecting generated and real images into pretrained feature spaces. Nonetheless, as shown by the results, we achieve further improvements by combining our method with ProjectedGAN, indicating that our method is orthogonal to ProjectedGAN because our FreGAN and ProjectedGAN are designed from different perspectives. More results and analysis of combining our method with ProjectedGAN will be presented in the next version of our paper.
> > >
> > > Hope that the above discussions can address your concerns. Thank you once more for your efforts and constructive suggestions.

---

> > > ### Comment · Reviewer_Xdtv · 2022-08-08
> > > **Response**
> > >
> > > Thank you for providing these additional experiments. Did I understand correctly that the numbers for ProjectedGAN were obtained by re-training the original model? It is surprising to me that the numbers match exactly those provided in the paper, while the models in the ProjectedGAN were trained on > 1M images (e.g. 3.8M on AFHQ-Dog) and you're reporting results for training on 1M images?

---

> > > > ### Author Response · Authors · 2022-08-08
> > > > **Response to Reviewer Xdtv**
> > > >
> > > > Dear Reviewer Xdtv,
> > > >
> > > > Thank you a lot for the review and the response.
> > > >
> > > > The results of Table 1 on 100-shot and AnimalFace datasets are obtained by training the original ProjetedGAN and the version combined with our method for 1M images on 2 V100 GPUs.
> > > > And the ProjetedGAN's reaults inTable 2 are quoted from the original paper as you gave in the first round. We train our combined model on these datasets with the recommonded config except saving the snapshots every 80 kimg.
> > > >
> > > > ``` python train.py --outdir=./AFHQdog/ --cfg=fastgan --data=./datasets/AFHQdog512.zip --gpus=4 --batch=64 --mirror=1 --snap=20 --kimg=10000 ```
> > > >
> > > > Four  A100 GPUS are rented to perform these experiments, and the best FID results are reported (~7M for AFHQ and ~1.2M for Pokemon), FID is basically consistent or slightly fluctuated after that.
> > > > We will further perform more experiments to investigate the  compatibility  of our method.
> > > >
> > > > Sorry for the misunderstanding and please let us know if you wonder to know more details.
> > > >
> > > > Best, Authors.

---

> > > > > ### Comment · Reviewer_Xdtv · 2022-08-08
> > > > > **Response**
> > > > >
> > > > > Dear authors,
> > > > > thank you for clarifying this, it answers my remaining questions.

---

> > > > > > ### Author Response · Authors · 2022-08-08
> > > > > > **Response**
> > > > > >
> > > > > > Dear Reviewer Xdtv,
> > > > > >
> > > > > > It is our pleasure to address your concerns and thanks again for your comments and constructive suggestions.
> > > > > >
> > > > > > Best, Authors.

---

### Official Review · Reviewer_MrWM · 2022-07-08

**Rating:** 7
**Confidence:** 4
**Soundness:** 3 good
**Presentation:** 2 fair
**Contribution:** 3 good

**Summary:**

The paper proposes an effective approach to the problem of few-shot GAN learning by leveraging the frequency information from the generated images. Specifically, the author designs a model, FreGAN, which integrates wavelet transformation, a high-frequency discriminator (HFD), frequency skip connections (FSC), and high-frequency alignment (HFA) together. Experiments are conducted on multiple datasets with different number of samples. The ablation of the experiment well explains the functionality for each component.

**Questions:**

See weakness.

As a side note, I am also interested to know how the author would integrate the approach into StyleGAN. Would the high-frequency information be extracted from the modulation signal or the modulated feature maps?

**Limitations:**

Yes.

**Strengths And Weaknesses:**

Strengths:

1. The method is novel. To improve high-frequency components for low-shot GAN learning is interesting.
2. The comparisons are extensive. Both qualitative and quantitative comparisons show strong results.
3. The ablation study well explains the effectiveness for each component.

Weaknesses:

1. Using some visual examples/figures to illustrate the effectiveness of each component could be even better. For example, how does HFA work? As HFD already provides a supervision to enforce G to synthesize high-frequency component similar to the real images, why should we add additional constraint on the intermediate feature maps? Does HFA stabilize the training?
2. The clarity of the paper can be improved, for example, it is not clear WHAT $\mathcal{G}$ and $\mathcal{T}$ are in Eqn. (7).

---

> ### Author Response · Authors · 2022-08-02
> **Response to Reviewer MrWM**
>
> Thank you for the review. We highlighted the revised content in the revised version, please refer to. The Answers to specific questions:
>
> Q1: Use visual figures to illustrate the efficacy of each component.
>
> A1: Thanks. As suggested, we include the visualization results of adding each component to the baseline in Figure 7 (page 12) of the appendix. The results indicate that each component of our method contributes to the quality of the generated images, which is consistent with the quantitative results in Table 5 (page 8) of the main paper.
>
> Q2: How does HFA work as HFD already provides a supervision of frequency signals to G? Does HFA stabilizes the training?
>
> A2: Thanks. HFD can indeed provide a supervision of frequency signals to G, but we believe that this is insufficient to make G competitive with D. This is because D can learn the frequency information from both real and generated data, whereas G can only obtain arbitrary guidelines from D. As a result, we propose HFA, which uses high-frequency signals from D as a regularizer to guide G.The visualization results of ablation studies in Figure 7 (page 12) of the appendix and the quantitative results in Table 5 (page 8) of the main paper suggest that HFA contributes to the synthesis quality, and the discriminator outputs in Figure 12 (page 15) of the appendix demonstrate that our proposed techniques contribute to more stable and effective training.
>
> Q3: Improve clarity of this paper, e.g., $\mathcal{G}$ and $\mathcal{T}$ in Eq (7).
>
> A3: Thanks. We checked and revised our paper, and made the equations and descriptions more clear. In Eq (7), $\mathbf{f}$ represents the intermediate features of D. $\mathcal{G}$ and $\mathcal{T}$ denote the processing on the features $\mathbf{f}$ and the input images $x$. D is considered as an encoder, and small decoders are trained to reconstruct the intermediate features.
>
> Q4: Integrate with StyleGAN2. Where is the high-frequency information extracted?
>
> A4: Thanks. We are also interested in integrating our techniques with StyleGAN2, which will be our future work (due to time and computation constraints ), as well as the performance of large-scale datasets. We apply our proposed techniques to the modulated feature maps instead of modulation signals.
>
> Hope that the above discussions address your concerns. Thanks for your effort and constructive suggestions again.

---

### Meta-Review · Area_Chair_nXQH · 2022-08-30

**Recommendation:** Accept
**Confidence:** Less certain

**Metareview:**

The majority of reviewers voted for accept, and we had a 7 among the mix, which I think cancels out the 4. This paper proposes using a wavelet method to bias the GAN towards generating the proper frequency distribution, which helps especially in the low-data regime. Overall I think low-data data modeling is important, and although this approach doesn't seem extremely novel, it seems useful, and on balance the reviewers voted to accept. I agree.

**Award:**

No

---

### Decision · Program_Chairs · 2022-09-14

Accept